# Airway basal cells show regionally distinct potential to undergo metaplastic differentiation

Yizhuo Zhou[1,2†], Ying Yang[1,3†#§], Lihao Guo[4†], Jun Qian[1,2], Jian Ge[1], Debora Sinner[5], Hongxu Ding[4*], Andrea Califano[6*], Wellington V Cardoso[1,2*#]

[1]Columbia Center for Human Development, Columbia University Irving Medical Center, New York, United States; [2]Department of Medicine, Pulmonary Allergy Critical Care, Columbia University Irving Medical Center, New York, United States; [3]Department of Genetics and Development, Columbia University Irving Medical Center, New York, United States; [4]Department of Pharmacy Practice and Science, College of Pharmacy, University of Arizona, Tucson, United States; [5]Neonatology and Pulmonary Biology Perinatal Institute, Cincinnati Children's Hospital Medical Center and University of Cincinnati, College of Medicine, Cincinnati, United States; [6]Departments of Systems Biology, Biochemistry & Molecular Biophysics, Biomedical Informatics, Medicine; JP Sulzberger Columbia Genome Center; Herbert Irving Comprehensive Cancer Center, Columbia University Irving Medical Center, New York, United States

*For correspondence:
hongxuding@arizona.edu (HD);
ac2248@cumc.columbia.edu
(AC);
wvc2104@cumc.columbia.edu
(WVC)

†These authors contributed
equally to this work

Present address: §Program
in Epithelial Biology and
Department of Dermatology,
Stanford University School of
Medicine, Stanford, United
States

#Lead contact

Competing interest: The authors
declare that no competing
interests exist.

Reviewing Editor: Edward
E Morrisey, University of
Pennsylvania, United States

**Abstract** Basal cells are multipotent stem cells of a variety of organs, including the respiratory tract, where they are major components of the airway epithelium. However, it remains unclear how diverse basal cells are and how distinct subpopulations respond to airway challenges. Using single cell RNA-sequencing and functional approaches, we report a significant and previously underappreciated degree of heterogeneity in the basal cell pool, leading to identification of six subpopulations in the adult murine trachea. Among these, we found two major subpopulations, collectively comprising the most uncommitted of all the pools, but with distinct gene expression signatures. Notably, these occupy distinct ventral and dorsal tracheal niches and differ in their ability to self-renew and initiate a program of differentiation in response to environmental perturbations in primary cultures and in mouse injury models in vivo. We found that such heterogeneity is acquired prenatally, when the basal cell pool and local niches are still being established, and depends on the integrity of these niches, as supported by the altered basal cell phenotype of tracheal cartilage-deficient mouse mutants. Finally, we show that features that distinguish these progenitor subpopulations in murine airways are conserved in humans. Together, the data provide novel insights into the origin and impact of basal cell heterogeneity on the establishment of regionally distinct responses of the airway epithelium during injury-repair and in disease conditions.

## Editor's evaluation

This study provides key evidence for basal cell heterogeneity in the airways of the lung. Critical evidence of transcriptional and signaling pathways that regulate the spatial patterning of ball cell heterogeneity are revealed. This information will hopefully guide a new understanding of how basal cells in the lung airways drive metaplastic diseases including cancers.

## Introduction

Basal cells (BCs) are tissue-specific adult multipotent stem cells of various organs, including skin, esophagus, olfactory, and the airway epithelia (*Rock et al., 2010*; *McKeon, 2004*). In the respiratory tract, BCs are found largely as a continuous layer of cells at the base of the pseudostratified epithelium of extrapulmonary airways in mice, and extending to small intrapulmonary conducting airways in humans (*Rock et al., 2010*; *Rock et al., 2009*). A large body of studies, over decades, has shown that BCs are essential for the homeostatic maintenance and repair of the respiratory epithelium. Dysregulated airway BC activities by environmental exposures and repeated injuries are known to result in aberrant self-renewal and epithelial differentiation, often associated with chronic obstructive pulmonary diseases (COPD) and pre-neoplastic lesions that can lead to lung squamous cell carcinoma (*Rock et al., 2010*; *Shaykhiev, 2021*; *Hynds and Janes, 2017*; *Ferone et al., 2020*).

The abundance and crucial role of these cells in the airway epithelium have, for many years, raised questions about whether BCs may represent a homogenous or a molecularly diverse pool of stem cells. Initial clues suggesting BC heterogeneity came from studies showing that they could differ in their ability to retain label in bromodeoxyuridine (BrdU) incorporation assays, and exhibit distinct clonogenic behavior due to different ability to activate the *Krt5* promoter (*Borthwick et al., 2001*; *Schoch et al., 2004*). This hypothesis was further strengthened by the identification of a subpopulation of *Krt14* +BCs, shown by lineage tracing to rapidly expand to generate additional BCs but not differentiated components of the damaged airway epithelium post-injury (*Hong et al., 2004*; *Ghosh et al., 2011*). Long-term clonal analysis using a *Krt5*$^{CreERT2}$ mouse line in combination with mathematical modeling, suggested that airway BCs comprise two subpopulations of multipotent stem cells. These were present in equal numbers and differed in their commitment from a *bona fide* stem cell to an intermediate luminal progenitor, not particularly associated with a specific cell niche (*Watson et al., 2015*). Mouse models of lung injury-repair have shown that, immediately after severe injury of the airway epithelium and prior to regeneration of the luminal cells, two discrete BC subpopulations can be identified and distinguished by expression of determinants of cell fate commitment to the secretory or ciliated cell lineage (*Pardo-Saganta et al., 2015*).

Single cell RNA sequencing (scRNA-Seq) analysis and computational modeling approaches have provided new insights into BC diversity in the adult lung and are helping describe the differences in BC subpopulations as the manifestation of discrete, co-existing cell states. One such study reports two BC states in the adult human lung, based on the degree of maturation of these cells, distinguished by the expression levels of TRP63 and NPPC (natriuretic peptide C) (*Vieira Braga et al., 2019*). Another study, also in human lungs, stratifies BCs into multipotent, proliferating, primed secretory, and activated subpopulations (*Carraro et al., 2020*). A third study, reports human BCs as proliferating, differentiating, or quiescent, and shows differential spatial enrichment of these populations, with proliferating and differentiating BCs present only in large airways, while quiescent BCs populate both large and small airways (*Travaglini et al., 2020*).

In mice, a combination of scRNA-Seq profiling and lineage tracing assays provided relevant information about cellular hierarchies and BC trajectories during differentiation but did not report on the diversity and distribution of BC subtypes in the tracheal epithelium (*Plasschaert et al., 2018*; *Montoro et al., 2018*). Using a sampling criterion based primarily on their spatial distribution, distinct BC phenotypes have been identified in tissues surgically dissected from the dorsal and ventral mouse trachea (*Tadokoro et al., 2021*).

Taken together, these observations support the notion that the basal stem cell pool is diverse, while leaving key unresolved questions. What is the actual spectrum of BC diversity in a resting, unperturbed state of the adult murine trachea? Does BC diversity translate into distinct abilities to regenerate or undergo aberrant repair in injured airways? Is there any evidence of conservation in these features between mouse and human airways?

To address these questions, we used an unbiased scRNA-Seq approach to identify components of the BC pool in adult *Trp63*$^{CreERT2}$; *Rosa26*$^{lox-STOP-lox-tdtomato}$ mice and examined their differences in functional in vitro assays and in vivo models of injury-repair. Our studies show a previously underappreciated BC heterogeneity in the adult murine airway, with six molecularly distinct BC subtypes. Among these, we identified two major subpopulations characterized by their abundance, unique signature, and regional distribution in the tracheal epithelium. Functional analyses showed that these two subpopulations responded differently to perturbations when isolated and cultured as organoids, and

during repopulation of the airway epithelium post-injury in intact animals. Finally, we found that BC heterogeneity is established during embryonic development and showed that key features of murine BCs are conserved in BC from human airways.

## Results

### scRNA-Seq reveals six distinct BC subpopulations in adult murine trachea under homeostasis

Mouse models of lung injury have provided major insights into how BCs contribute to regenerating the airway epithelium. Nevertheless, there is still limited information on how diverse the BC stem cell pool is in the unperturbed adult murine trachea. Although previous studies using scRNA-Seq approach provided clear evidence of BC heterogeneity, methodological differences, including choice of mouse lines for BC isolation, the ability to capture all subpopulations, and choice of computational analysis, have not allowed a full understanding of the extent to which steady-state BCs differ (*Plasschaert et al., 2018*; *Montoro et al., 2018*).

To have a more precise idea about the composition of the BC pool, we specifically sorted these cells using a *Trp63*$^{CreERT2}$; *Rosa26*$^{lox-STOP-lox-tdtomato}$ mouse line known to label universally all BCs, thus providing a greater chance to identify all subtypes. This choice was supported by extensive evidence of *Trp63* expression in BCs throughout the body and the inability of these cells to form in the absence of *Trp63* (*Daniely et al., 2004*; *Romano et al., 2012*; *Fletcher et al., 2011*; *Zhao et al., 2014*; *Yang et al., 1999*). Adult *Trp63*$^{CreERT2}$; *Rosa26*$^{lox-STOP-lox-tdtomato}$ mice were administered tamoxifen (240 µg/g body weight) and examined 72 hr later. Co-staining of TRP63 + cells with tdTomato in tracheal sections of these mice confirmed efficient labeling of all BCs (*Figure 1A–B*). KRT5 overlapped with TRP63 and tdTomato, and only rare KRT5 +TRP63 and tdTom- suprabasal cells could be identified, as previously reported (*Salahudeen et al., 2020*; *Mori et al., 2015* ; *Figure 1B*).

Thus, we sorted tdTom$^+$ EPCAM$^+$ Lin$^-$ CD104$^{HI}$ airway BCs, validated their purity using cytospin (*Figure 1—figure supplement 1A-B*)**,** and generated scRNA-Seq profiles using a 10× Genomics Chromium platform. A total of 4209 single cells passed quality control, with median values of 10,802 transcripts and 2956 genes per cell detected (see Methods). Using the Cellranger toolkit (10× Genomics), followed by marker gene annotation, we identified six distinct BC subpopulations, which were confirmed to originate from the respiratory lineage (*Epcam +Nkx2-1+*) and to express established adult BC markers (*Trp63, Krt5, Aqp5, Aqp3,* and *Pdpn*) (*Figure 1—figure supplement 1C*).

Cluster analysis showed that these subpopulations differed widely in their contribution to the BC pool. Notably, two clusters were prominently represented, accounting for 85.8% of all sequenced BCs. They were found at an approximate 1:1 ratio and were named as BC-1 (46.1%) and BC-2 (39.7%). Both lacked *Krt14* expression, while expressing comparable levels of canonical BC markers (*Trp63*, *Krt5*, and *Krt15*). The remaining four clusters comprised only a small fraction of the total BC pool (14.2%). Based on their gene signatures, featuring markers associated with differentiated cell phenotypes, these were classified as Basal-proliferating (Pro.; *Mki67, Ccna2, Aurkb, Ube2c*: 2.1%), Basal-secretory (Sec.; *Scgb1a1, Scgb3a1, Scgb3a2, Muc5b*: 5.8%), Basal-squamous (Sq.; *Krt13, Krt4, Krt6a, Sprr2a3*: 3.3%), and a cluster represented by expression of genes associated with mesenchymal cells (Mes.; Basal-mesenchyme-like), which accounted for 3.1% of all profiled cells (*Figure 1C–D*, *Figure 1— figure supplements 1 and 2*). This scRNA-Seq dataset can be explored through MmTrBC data portal at http://visualify.pharmacy.arizona.edu/MmTrBC/.

A closer look at BC-1 and BC-2 showed that, in contrast to the other BC subpopulations, they were not characterized by enrichment in genes associated with any of the differentiated cellular phenotypes. If present, these genes were detected at very low levels and only in few cells of the cluster. Although not enriched for specific stem cell markers, BC-1 and BC-2 appeared to be the most stem-like of all the BC subpopulations. Moreover, their signatures did not seem to overlap with any of the BC subpopulations reported by previous scRNA-Seq studies in mice (*Figure 1C–D*, *Figure 1—figure supplements 1–3*).

### The two major BC subpopulations reside in distinct niches

Since BC-1 and BC-2 constituted the majority of the BCs in the airway epithelium, we further examined their transcriptional signatures to assess their differences, including against other BC subpopulations.

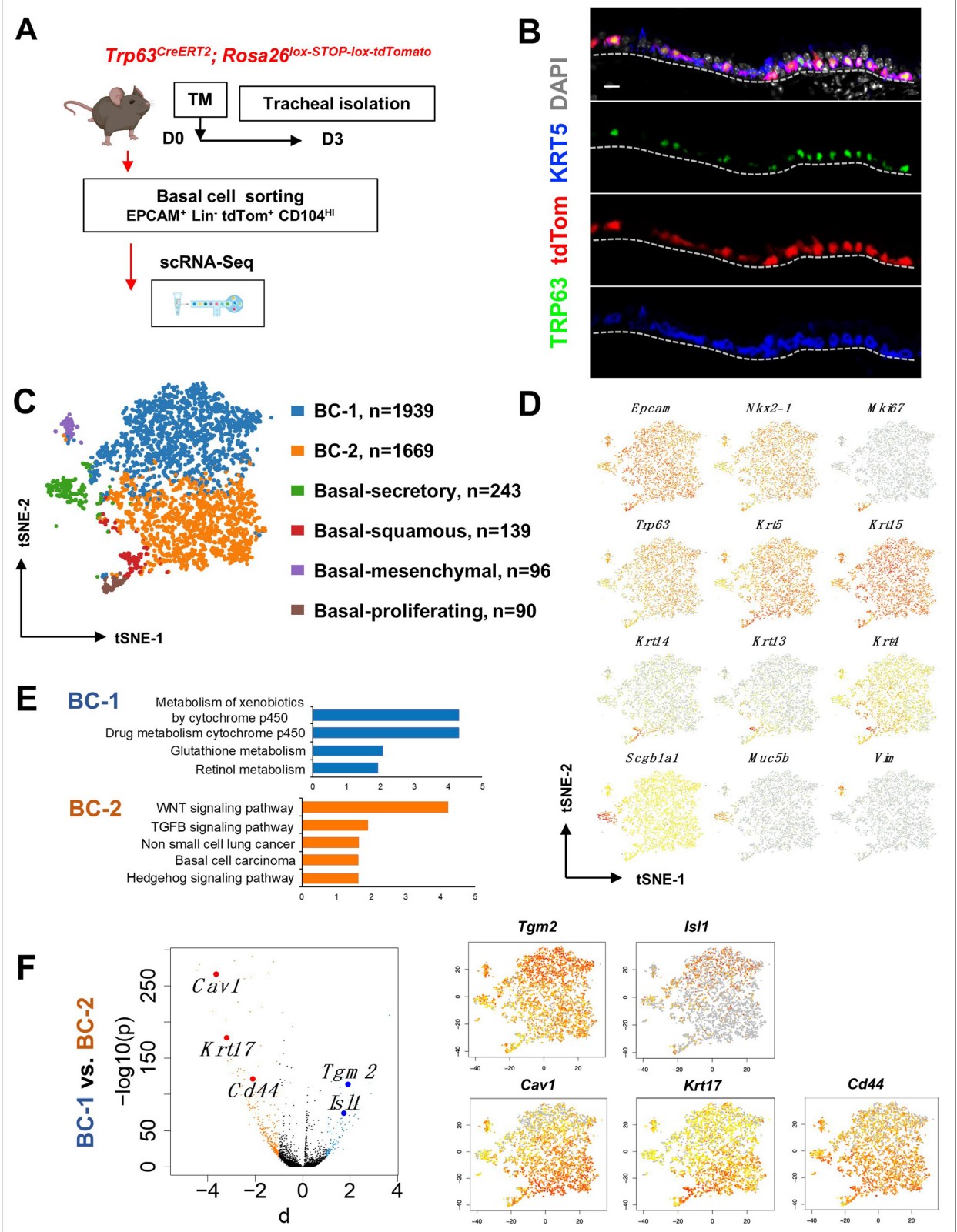

**Figure 1.** Single cell RNA sequencing (scRNA-Seq) reveals six distinct airway basal cell (BC) subpopulations in the murine trachea. (**A**) Diagram: strategy for labeling, isolation and scRNA-Seq analysis of BCs from *Trp63*^CreERT2^; *Rosa26*^lox-STOP-lox-tdTomato^ adult tracheas. (**B**) Immunofluorescence (IF) of tracheal sections showing efficient tdTom co-labeling with TRP63 +KRT5+BCs restricted to the basal layer of the airway epithelium. (**C**) tSNE visualization of the six airway BC subpopulations identified by scRNA-Seq, colored by cluster assignment of each population annotated using established lineage-

*Figure 1 continued on next page*

*Figure 1 continued*

specific or proliferation markers. (**D**) tSNE visualization of airway BC scRNA-Seq, colored by expression (log2(TPM +1)) of representative marker genes of the different BC subpopulations. (**E**) Enriched KEGG gene sets in BC-1 and BC-2 subpopulations. (**F**) Left panel: Volcano plot showing differentially expressed genes in BC-1 and BC-2 (right; black dots); genes significantly enriched in BC-1 were colored in blue (-log10(p)>10; d: average expression difference >1); those significantly enriched in BC-2 were colored in orange (-log10(q)>10; d < −1). Right panel: tSNE visualization of *Tgm2, Isl1, Cav1, Krt17, and Cd44* distribution in the BC subpopulations. Scale bar in (**B**): 10 μm.

The online version of this article includes the following figure supplement(s) for figure 1:

**Figure supplement 1.** scRNA-Seq of sorted mouse tracheal BCs reveals six distinct subpopulations.

**Figure supplement 2.** Mouse tracheal BC subpopulations exhibit distinct gene expression patterns.

**Figure supplement 3.** tSNE visualization of airway basal cell (BC) single cell RNA sequencing (scRNA-Seq) for BC markers identified from previous publications in mice and humans, colored by expression [log2(TPM +1)] of each marker gene.

**Figure supplement 4.** Basal cell (BC) subtypes showing unique gene expression signatures.

**Figure supplement 5.** Basal-squamous cluster showing unique gene expression signature compared to the other five basal cell (BC) clusters.

Despite some similarities, when compared to the other BC subpopulations, BC-1 and BC-2 had markedly distinct gene expression signatures (*Figure 1—figure supplements 1–2*, 4). Gene set enrichment analysis of KEGG pathways revealed enrichment of BC-1 in genes associated with drug metabolism, glutathione, and retinol metabolism. By contrast, BC-2 was differentially enriched in genes from the WNT, TGFb, and Hedgehog signaling pathways, as well as signatures associated with basal cell carcinoma and non-small cell lung cancer (*Shin et al., 2011*; *Youssef et al., 2012*; *Chen et al., 2014*; *White and Lowry, 2015*; *Figure 1E*).

We identified the most differentially expressed genes between BC-1 and BC-2 (-Log10(p)≥10; d (average expression difference)>1) to characterize their expression pattern and examine whether they could identify where these two subpopulations resided in the tracheal epithelium (*Figure 1F*). Candidates were selected based on statistical significance and availability of validated antibodies to confirm that these genes were not only transcribed but also translated. Based on these criteria, immunofluorescence (IF) was performed in sections of adult trachea to map the expression of *Tgm2* (transglutaminase-2, C polypeptide), *Isl1* (ISL LIM homeobox), *Cav1* (caveolin1), *Cd44* (Cd44 antigen, or Pgp1), and *Krt17* (keratin-17, K17) (*Figure 1F*). Co-IF with a KRT5 antibody confirmed the expression of these markers in BCs in the upper and lower trachea with no clear evidence of preferential distribution along the anterior-posterior axis (*Figure 2A–B*). However, a more detailed analysis of these markers in longitudinal and cross sections of adult tracheas showed a remarkable enrichment of TGM2 and ISL1, markers of BC-1, in the KRT5 + cells of the ventral epithelium (*Figure 2B*, Figure S6). This contrasted with the weak signals detected in BCs of the dorsal epithelium. Dorsal BCs were rather characterized by strong expression of CD44, CAV1, and KRT17, which in turn, were only weakly detected in ventral BCs (*Figure 2B*, *Figure 2—figure supplement 1*). Quantitative analysis of fluorescence intensity confirmed the significant differences in dorsal-ventral distribution of BC-1 and BC-2 markers (*Figure 2C*).

We performed similar IF analysis with representative markers of the other four BC subpopulations (Basal-secretory/Sec., Basal-squamous/Sq., Basal-proliferative/Pro., and Basal-mesenchymal-like/Mes.) as an attempt to identify where these BCs resided. Interestingly, this showed only isolated or rare groups of cells scattered throughout the airway epithelium with no consistent spatial distribution, unlike the pattern seen for BC-1 and BC-2 (*Figure 2—figure supplement 2*).

## BC-1 and BC-2 are functionally different and maintain their distinct ability to self-renew in vitro

To further investigate the phenotypic differences between BC-1 and BC-2 cells, we asked whether their unique gene signatures depended on active signals from their respective local microenvironments in the trachea. Given the major differences in tissue structure and gene networks between the dorsal and ventral tracheal mesenchyme, we examined the extent to which BC-1 and BC-2 preserved their identities if isolated and expanded in culture. For this, whole tracheas from adult mice had their membranous (dorsal) and cartilaginous (ventral) regions surgically dissected, and their respective BC subpopulations were isolated and cultured separately. Approximately equal amounts of cells ($3.3 \times 10^4$)

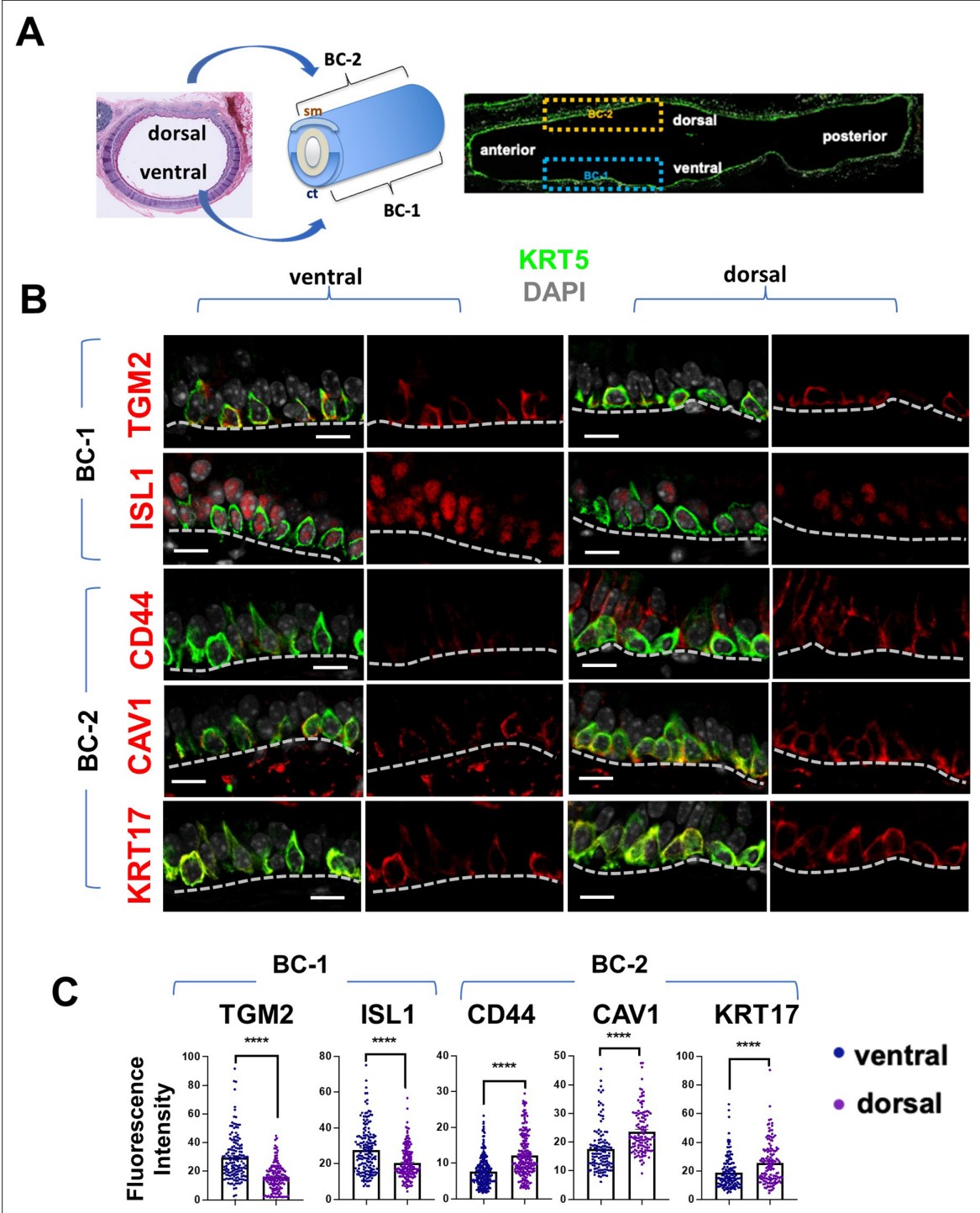

**Figure 2.** BC-1 and BC-2 reside in distinct niches of the adult tracheal epithelium. (**A**) Schematic representation of the differential distribution of BC-1 and BC-2 markers in basal cells along the dorsal-ventral axis of the adult trachea (diagram: sm, smooth muscle; ct, cartilage). (**B**) Immunofluorescence of BC-1 and BC-2 markers co-labeled with KRT5 in serial sections of the adult trachea showing differential enrichment in ventral or dorsal epithelium, respectively. (**C**) Quantification of the fluorescence intensity of BC-1 and BC-2 markers confirms the significant differences in expression between ventral and dorsal BCs. Graphs: Each dot represents the average fluorescence intensity value of a single KRT5 + cell analyzed (already clarified in methods: Fluorescence intensity and area (μm²) for each cell was measured by Zen 2.3 lite software.); bars represent the mean ± SEM of 8–15 paired fields counted from 5 to 9 sections; n=3 animals (See also Figure S6). Student's t-test, ****p<0.0001. Scale bars, 10 μm.

The online version of this article includes the following figure supplement(s) for figure 2:

**Figure supplement 1.** BC-1 and BC-2 populations reside in regionally distinct niches of the adult tracheal epithelium.

*Figure 2 continued on next page*

*Figure 2 continued*

**Figure supplement 2.** Basal cell (BC) subpopulations other than BC-1 and BC-2 are relatively rare and have no consistent spatial distribution in the adult tracheal epithelium.

were seeded in Transwell plates and expanded under submerged conditions for 7 days, as previously described (*Mori et al., 2015*).

No obvious difference in growth was observed between BCs from the two regions as they expanded, and both reached confluency by the first week in culture (*Figure 3A*). Double-IF of the BC-1 and BC-2 markers with KRT5 confirmed expression in cultured BCs from both groups. However, TGM2 and ISL1 signals were distinctly stronger in ventral BC cultures, compared with dorsal cultures. This contrasted with the stronger expression of CAV1, CD44, and KRT17 in dorsal BC cultures (*Figure 3A*). q-PCR analysis of these cultures showed differences overall consistent with the in vitro and in vivo IF patterns (*Figure 3B*). These observations support the hypothesis that, BC-1 and BC-2 cells isolated from the adult trachea can maintain their phenotypes in vitro, independent of their respective microenvironments for at least a week, thus providing an opportunity to assess their functional differences as adult progenitors of the airway epithelium.

To identify differences in progenitor cell behavior, BC-1 and BC-2 cells were isolated from ventral and dorsal trachea as above, and $3.0 \times 10^3$ cells from each region were embedded in Matrigel and cultured in Transwell plates to form epithelial-only organoids (*Figure 3C*, *Figure 3—figure supplement 1*; *Rock et al., 2009*). Analysis of these cultures over a 2 weeks period confirmed the ability of both subpopulations to form large-size (≥35.65 μm) and small-size (<35.65 μm) tracheospheres at equal efficiency. However, organoids derived from the dorsal (BC-2) cultures were found at a significantly higher number compared those derived from ventral (BC-1) cells (*Figure 3C*; 10.41 ± 0.009% vs 7.32 ± 0.006%, respectively). The higher colony-forming efficiency of dorsal (BC-2)-derived organoids was consistent with observations from a previous study comparing ventral and dorsal BCs in similar clonogenic assays (*Tadokoro et al., 2021*).

Taken together, these data suggest that the distinct transcriptional signatures of BC-1 and BC-2 are associated with functional differences in their ability to self-renew and presumably differentiate in vitro and in vivo.

## BC-2 cells are enriched in regulators of a squamous differentiation program but are not Basal-squamous

To identify differentially expressed genes that could account for the functional differences in progenitor cell behavior between BC-1 and BC-2, we revisited their gene expression signatures, searching for potential candidate regulators (*Figure 1—figure supplement 4*). This revealed a significant enrichment in genes associated with a squamous cell differentiation program. Notably, we found key transcription factors differentially expressed and upregulated in BC-2, including the Krüppel-like Factors *Klf4*, *Klf6*, and *Klf13*, *Barx2*, and AP1 family members (*Junb*, *Jund*, *Atf3*) (*Figure 1—figure supplement 4B*), all reported to promote squamous cell commitment in stratified epithelia (*Araya et al., 2007*; *Bialkowska et al., 2017*; *Sastre-Perona et al., 2019*; *He et al., 2015*; *Olson et al., 2005*; *Chen et al., 2018*; *Ding et al., 2013*; *Eckert et al., 2013*). The BC-2 signature also differed in its enrichment in genes from the Hedgehog/TGFb/WNT pathways involved in squamous differentiation (*Figure 1E–F*, *Figure 1—figure supplement 4A-B*). Moreover, GSEA revealed strong enrichment of a smoking-related squamous metaplasia signature in BC-2, compared to BC-1 cells (*Goldfarbmuren et al., 2020*: *Figure 1—figure supplement 4C*).

BC-mediated generation of a squamous stratified epithelium is a key event during differentiation of the skin and esophagus. In the lung, however, it is a hallmark of an aberrant program triggered by sustained epithelial injury giving rise to squamous metaplasia, a reversible lesion that can also lead to pre-malignant conditions (*Araya et al., 2007*; *Ooi et al., 2014*; *Auerbach et al., 1961*).

Our cluster analysis identified a small subpopulation of Basal-squamous cells characterized by a distinctive *Krt13*, *Krt4*, *Krt6a*, and *Sprr2a3* marker gene signature, also characteristic of differentiated squamous cells (*Figure 1C–D*, *Figure 1—figure supplements 1C and 2*, 5). Importantly, none of these marker genes were found in the list of the 100 most differentially-expressed genes in BC-2 compared to BC-1, in spite of the BC-2 enrichment in regulators of the squamous cell differentiation program (*Figure 1—figure supplement 4*). Moreover, genes such as *Krt6a*, *Serpinb2*, and *Krt13*

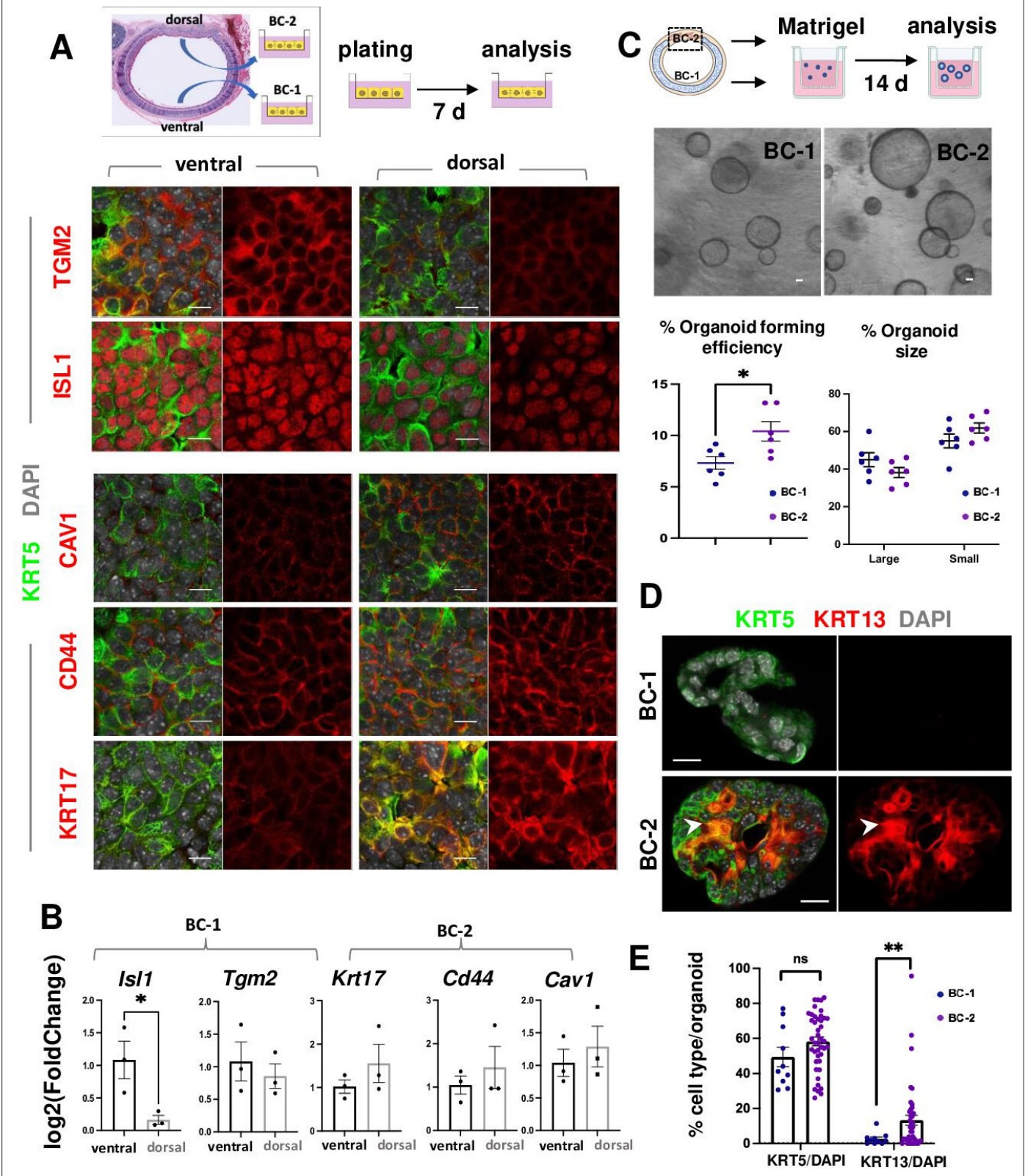

**Figure 3.** BC-1 and BC-2 differ in their ability to form organoids and activate a metaplastic program of differentiation in primary cultures. (**A**) Strategy for isolation, expansion and analysis of BC-1 and BC-2 in submerged cultures. IF at day 7 showing preserved differential enrichment of BC-1 and BC-2 markers in cultures derived from ventral and dorsal BCs, respectively. (**B**) qPCR of day 7 cultures depicting differences in marker gene expression [log2(FoldChange)] between these subpopulations. Bars are mean ± SEM of 3 replicate cultures. (**C**) Experimental approach for functional analysis of

*Figure 3 continued on next page*

*Figure 3 continued*

BC-1 (ventral) and BC-2-derived (dorsal) cultures in organoid assays. Bright field images of BC-1 and BC-2-derived 3D-organoids at day 12. Dot plots: (left): Organoid-forming efficiency (percentage of organoid number / total seeding basal cell number); (right) percentage of small and large-sized day 12 organoids derived from BC-1 or BC-2 progenitors. Graphs are mean ± SEM of values from 6 independent cultures from each group. See methods for quantification details. (**D**) IF of KRT5 and KRT13 in day 14 organoids showing similar KRT5 signals but abundant KRT13 signals in BC-2-derived cultures compared to those from BC-1. (**E**) Percentage of KRT5 +or KRT13 + cells in total DAPI-labeled cells from BC-1 or BC-2-derived organoids. Bars represent mean ± SEM of values in each group (number of organoids: BC-1: n=10, BC-2: n=43). Student's t-test, *p<0.05, **p<0.01, n.s., not significant. Scale bars:10μm.

The online version of this article includes the following figure supplement(s) for figure 3:

**Figure supplement 1.** Absence of mesenchymal contamination in organoid cultures.

(*Tadokoro et al., 2021*), previously reported in BCs isolated from the dorsal murine trachea, were among the genes differentially expressed in our Basal-squamous but not in the BC-2 cluster, which we found localized to the dorsal BCs (*Figure 1—figure supplement 5A-C*).

Thus, while largely uncommitted during homeostasis, BC-1 and BC-2 appear to have distinct differentiation potentials, based on their gene signatures. Our data are also consistent with the idea that, in spite of the potential of BC-2 to initiate a program of squamous differentiation, these cells are widely different from the Basal-squamous population not only in the signature but also in abundance and spatial distribution.

## BC-1 and BC-2 differ in their ability to respond to environmental perturbations in primary cultures

We speculated that BC-2 cells were more sensitive to environmental perturbations, and that subjecting BC-1 and BC-2 to similarly altered culture conditions could reveal differences in their differentiation potential. To explore this possibility, organoids were generated with BCs isolated from either ventral or dorsal regions, as they largely represented the BC-1 and BC-2 subpopulations, respectively (*Figure 3*). As previously described, after 7 days growing in a serum-based medium (MTEC Plus), culture conditions were changed by shifting them to a serum-free (MTEC/SF) medium and the culture period was extended for an additional week. Cultures were then analyzed for their ability to maintain their KRT5 +progenitor cell pool and to induce KRT13, a marker of noncornified squamous epithelium and known to mark initiation of a metaplastic program in the airway epithelium of mice, rats and humans (*Bolton et al., 2009*). *Krt13* was also chosen as it featured as a topmost upregulated gene in the Basal-squamous cluster and, thus, a key indicator of cells undergoing commitment to a squamous cell fate choice.

IF and quantitative analysis of KRT5 in these organoids showed no difference in the relative number of KRT5-labeled cells between groups (BC-1: 49.43 ± 5.54%, BC-2: 58.29 ± 2.50%, p=0.17), suggesting that the shift in culture conditions (serum-based to serum-free media) had a similar impact in the ability of BC-1 and BC-2 to self-renewal. By contrast, the number of KRT13 + cells was strikingly different (BC-1: 2.65 ± 1.08%, BC-2: 13.26 ± 2.86%, p=0.001) (*Figure 3D–E*), suggesting that the change/perturbation in the culture environment triggered a KRT13 program of differentiation in BC-2 not seen in BC-1 progenitors. This differential behavior was also observed when BC-1 and BC-2 cells were similarly expanded in serum-containing media and then allowed to differentiate as 2D air-liquid-interface (ALI) organotypic cultures in serum-free medium (see subsequently).

In the respiratory epithelium, squamous metaplasia is strongly linked to perturbations in the progenitor cell program triggered by disruption in the retinoic acid (RA) signaling. This has been extensively reported as a manifestation of vitamin-A deficiency in humans and in animal models, as well as in cells cultured in the presence of pharmacological antagonists of the RA pathway (*Lancillotti et al., 1992*; *Chopra, 1983*; *Choi et al., 2003*; *Nettesheim et al., 1990*). Given the differential enrichment of retinoid-related genes between BC-2 and BC-1 cells, we examined how these progenitors responded when cultured in the presence of the pan-retinoic acid receptor inhibitor BMS-493 (BMS).

IF analysis of KRT13 + cells at ALI day 7 showed nearly no labeling in control BC-1-derived cultures in contrast to BC-2 controls, which had significantly higher percentage of KRT13 + cells (1.54 ± 0.47% and 0.39 ± 0.39% in BC-2 and BC-1, respectively), and higher levels of *Krt13* expression shown by qRT-PCR (*Figure 4A–C*).

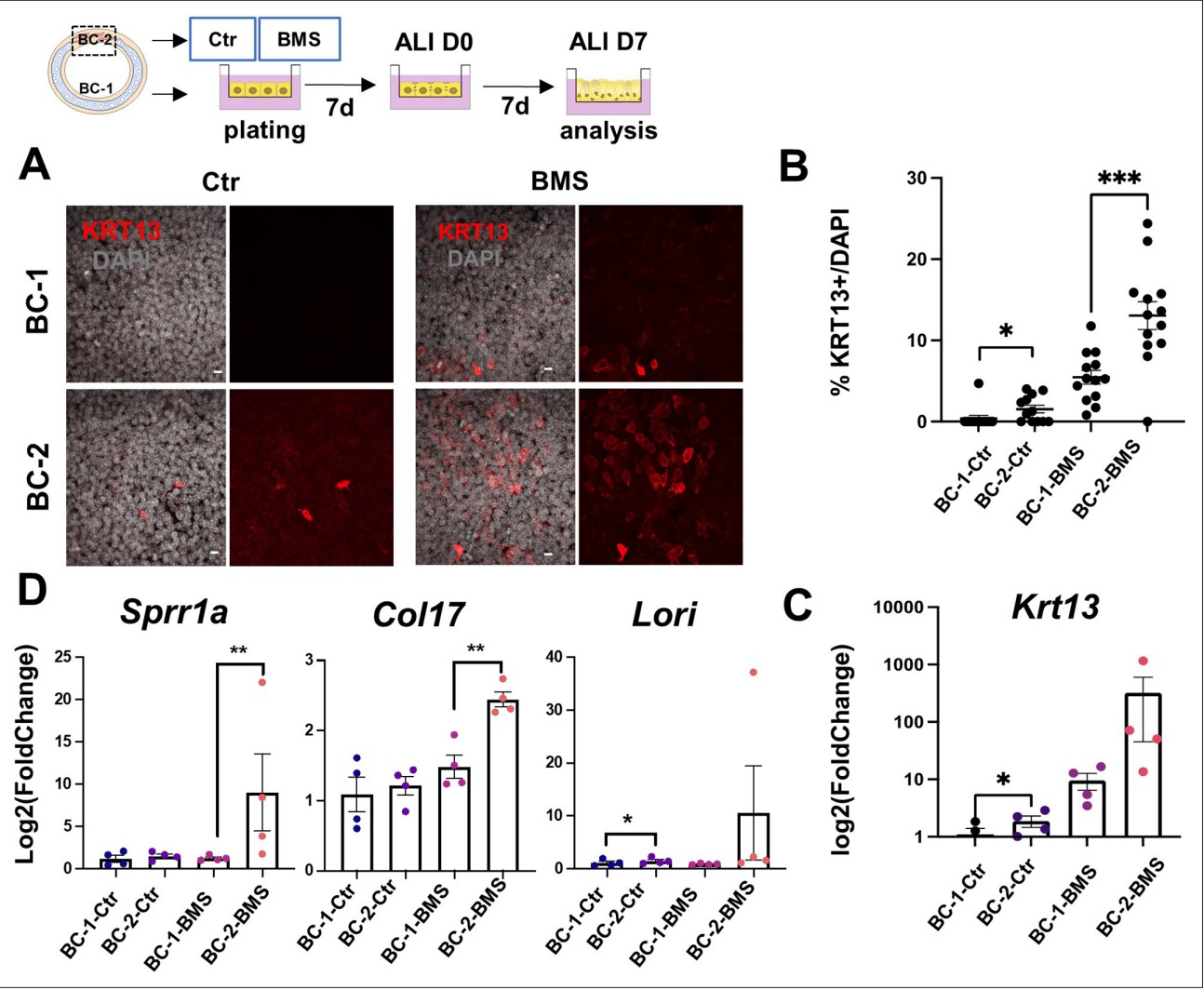

**Figure 4.** Differential induction of squamous-associated markers in retinoid-deficient BC-2-derived organotypic cultures. Expansion and differentiation of BC-1 and BC-2 populations isolated from ventral or dorsal trachea, respectively, cultured in air-liquid-interface (ALI) in control or RAR antagonist (BMS493) conditions. (**A**) Increased KRT13 IF labeling in BC-2-derived cultures compared to BC-1 in both control and BMS-treated conditions. Scale bars: 10 μm. (**B**) Morphometric analysis of KRT13 + cells in ALI day 7 control or BMS-treated cultures. Each dot represents the percentage of KRT13 + cells in total cell population per field. Graph: mean ± SEM of more than 4 fields per batch, n=3 batches for each condition. (**C–D**) qPCR of *Krt13, Sprr1a, Col17* and *Loricrin* in ALI day 7 control or BMS-treated cultures. Values [log2(foldchange)] are normalized to BC-1 controls. Bars are mean ± SE of n=4 replicates per culture condition. Student's t-test, *p<0.05, **p<0.01, ***p<0.001.

The online version of this article includes the following figure supplement(s) for figure 4:

**Figure supplement 1.** RAR signaling is active in ventral and dorsal luminal cells but not in basal cells (BCs) of the adult tracheal epithelium during homeostasis.

BMS treatment resulted in a marked increase in number of KRT13 + cells in both BC-1 and BC-2-derived cultures. However, RA signaling disruption had a much greater impact in the induction of KRT13 in BC-2-BMS than in BC-1-BMS-derived cultures (BC-1-BMS: 5.46 ± 0.85% vs BC-2-BMS: 13.06 ± 1.72%; *Figure 4A–C*). Remarkably, analysis of other gene markers of the program of squamous cell differentiation, besides *Krt13*, confirmed the differential induction in BMS-treated BC-2 cultures compared to BC-1 cultures (*Figure 4D*).

The response of both BC-1 and BC-2 to BMS was consistent with the idea that endogenous RA signaling is activated during expansion and differentiation in vitro and presumably differentially required to regulate these events in BC-1 vs BC-2 cells. This raised the question whether RA signaling was differentially active in these two BC subpopulations in the intact adult trachea under homeostatic

conditions. To address this issue, we used a RARE-LacZ reporter mice (*Rossant et al., 1991*) extensively shown to label cells activating RA signaling in various tissues and contexts. We compared the distribution of LacZ signals in the ventral and dorsal epithelium in tracheal sections of adult mutants co-stained with X-gal and markers of basal, secretory and multiciliated cells. Surprisingly, LacZ signals were clearly present in luminal cells of both ventral and dorsal epithelium but not in BCs (*Figure 4— figure supplement 1*). This suggested that, under homeostatic conditions, BCs are unlikely to activate RA signaling. However, the presence of components of the RA biosynthetic pathway in BCs suggests that they may produce RA or its precursors making available to neighbor luminal cells to drive or maintain their differentiation program. Consistent with this, there is evidence that skin BCs (keratinocytes) metabolize retinoid precursors and serve as a source of retinol for differentiation of the upper skin layers (*Kurlandsky et al., 1996*). Interestingly, BC-1 differs from BC-2 in its enrichment in Retinol metabolism genes, suggesting that BC-1 could be better equipped of producing RA compared with BC-2. This could presumably contribute to generate the dorsal-ventral differences in sensitiveness of the tracheal epithelium to an RA-deficient environment. Further studies are required to proper explore this speculation.

## BC-2 shows markedly distinct behavior during initiation of repair of the damaged epithelium in mouse models of injury

These intriguing observations led us to inquire whether severe injury of the airway epithelium in the intact animal could reveal intrinsic differences in the BC-1 vs BC-2-mediated repair programs. For this, we used two established mouse models of lung injury-repair in vivo. Intraperitoneal injection of Naphthalene, a byproduct of the cigarette smoke metabolized by cytochrome P450 enzymes results in massive injury and sloughing of club cells of the airway epithelium. In the trachea, repair is subsequently initiated by activation of a BC program of regeneration that repopulates the epithelium within 2–3 weeks (*Tata et al., 2018*). Polidocanol is a detergent/sclerosing agent that induces extensive and severe epithelial sloughing when injected intratracheally in mice, also eliciting a BC-mediated regenerative response that reconstitutes the airway epithelium (*Paul et al., 2014*).

Injury was induced with Naphthalene (Nap) or Polidocanol (Poli) in 8 week-old adult mice and animals were euthanized during active epithelial repopulation (Nap: 5dpi; Poli: 7dpi) and later (both: 15dpi), once regeneration was well advanced in the tracheal epithelium. IF was performed in tracheal sections oriented for simultaneous exposure of both the ventral and dorsal tracheal epithelium, based on their association with cartilage or a smooth muscle layer. We compared the dynamics of repopulation between the ventral and dorsal epithelium using proliferation (KI67) and cell-type specific markers (secretory: CC10, SCGB3A2; multiciliated: FOXJ1; basal: KRT5, TRP63; and squamous: KRT13). Control (uninjured) animals (0dpi) showed a balanced distribution of these phenotypes with no significant differential enrichment between dorsal and ventral cells and only rare KRT13 + cells, also without preferential dorsal-ventral distribution (*Figure 5A–C and H*, *Figure 5—figure supplement 1A,3C-D*).

Unexpectedly, analysis of the earlier stages of regeneration in both Naphthalene and Polidocanol injury models revealed major differences in the kinetics of repopulation between the dorsal and ventral trachea. While the ventral epithelium of 5dpi-Nap and 7dpi-Poli tracheas was already populated by a large number of secretory and multiciliated cells, these profiles were still missing or greatly diminished in the dorsal side (*Figure 5E and G*, *Figure 5—figure supplement 1B-C*). Differences were particularly prominent in the Polidocanol model, as seen by the significantly lower number of dorsal CC10+, SCGB3A2+ and FOXJ1 + cells (*Figure 5H*). Analysis of cell proliferation showed no dorsal-ventral difference in KI67 labeling in BCs from uninjured control tracheas. KI67 labeling was increased in BCs during repair post-injury and was disproportionally higher in dorsal BCs from 7dpi-Poli tracheas (*Figure 5—figure supplement 1*).

In addition, the dorsal epithelium showed a striking induction of KRT13 expression in KRT5 + cells and a massive increase in KRT13 + cells in the nascent regenerating luminal cells (*Figure 5*). This was consistent with our findings from the perturbation assays in BC-2 (dorsal)-derived organoids and ALI (*Figures 3D and 4A–C*). Although also induced in the ventral epithelium, KRT13 labeling occurred in a significantly smaller scale and allowed regeneration to proceed. Indeed, in 5dpi-Nap mice KRT13 +labeling in the tracheal epithelium increased dorsally from 0.71 ± 0.37% (control) to 45.47 ± 4.74%, while ventrally labeling increased from 0.2 ± 0.12% (control) to 8.36 ± 2.55%. Similarly, in 7dpi-Poli KRT13 +comprised 53.41 ± 5.33% of all the dorsal regenerating epithelium, compared

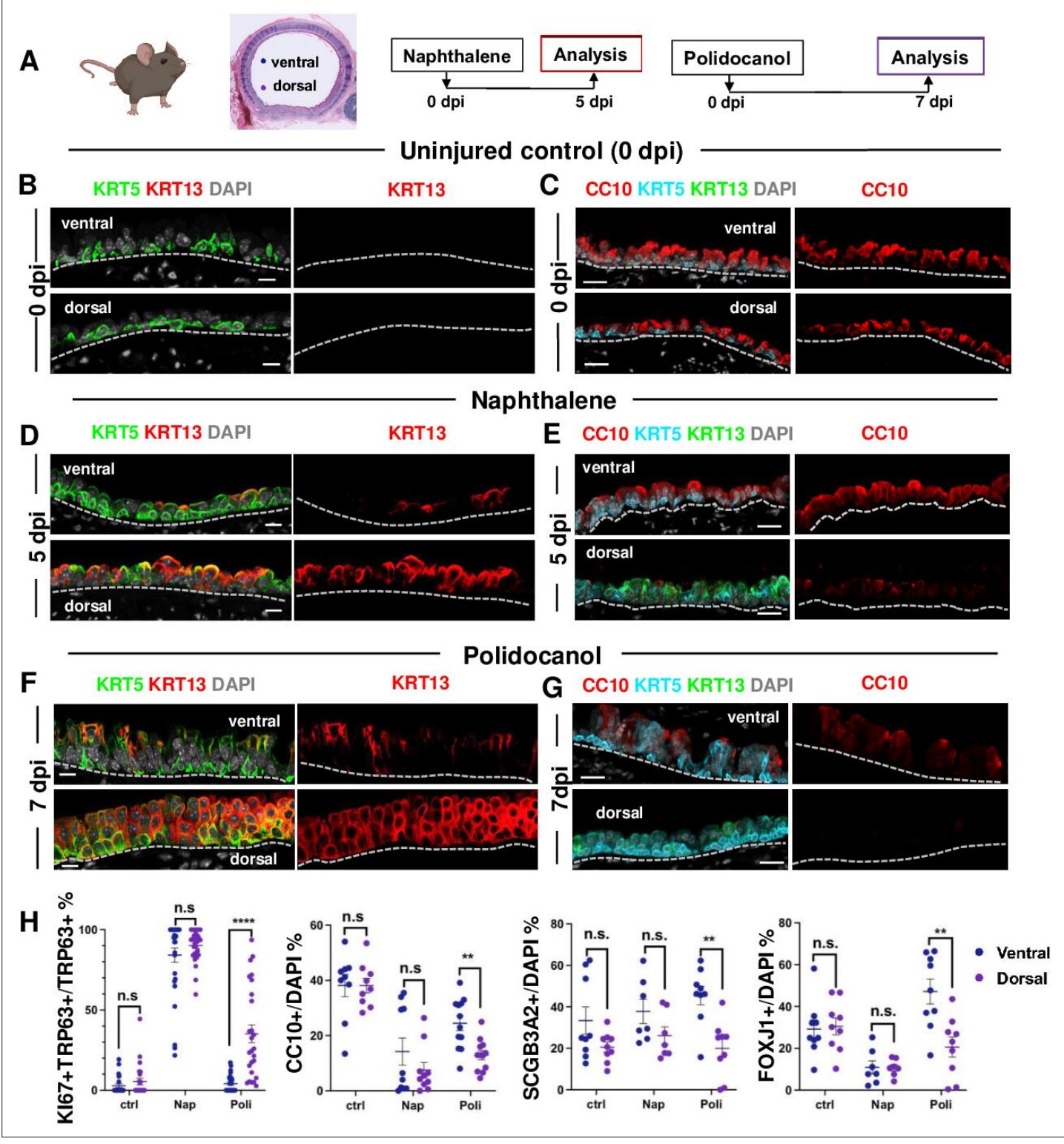

**Figure 5.** BC-2 and BC-1 have distinct behaviors during initiation of repair of the damaged epithelium in mouse models of injury. (**A**) Diagram: strategy for induction of airway injury by Naphthalene (Nap, intraperitoneal) or Polidocanol (Poli, intratracheal) and analysis at 5dpi post-Nap or 7dpi post-Poli. (**B–C**) IF of tracheal sections from adult uninjured animals (0dpi, uninjured control, n=4 mice per group) showing typical dorsal-ventral distribution of KRT5 and CC10 (no KRT13 signals) in the basal and luminal layers, respectively. (**D–G**) Markedly distinct kinetics of repopulation of the airway epithelium by BCs from the ventral (BC-1) compared to dorsal (BC-2) regions at 5dpi (Naphthalene) or 7dpi (Polidocanol) (n=3 mice per group): extensive KRT13 expression in KRT5 + and luminal cells of the dorsal epithelium and low CC10 expression. (**H**) Morphometric analysis: percentage

*Figure 5 continued on next page*

*Figure 5 continued*

of proliferating (KI67+) BCs, and luminal (CC10+, SCGB3A2+, FOXJ1+) cells in ventral or dorsal epithelium of 5dpi or 7dpi (see also Figure S10, S11). For KI67 +TRP63+/TRP63+% (far left panel), each dot represents the percentage of proliferating BCs (KI67 +TRP63+) in total BC population (TRP63+) from each view. Graphs: mean ± SEM of 2–20 views per animal; 0dpi-control: n=4 animals; 5dpi-Nap: n=3 animals; 7dpi-Poli: n=3 animals. For CC10+/DAPI%, SCGB3A2+/DAPI%, FOXJ1+/DAPI% (right three panels), each dot represents the percentage of each lineage-committed cells in total epithelial population from each view. Graphs: mean ± SEM of 1–5 views per animal; 0dpi-control: n=4 animals; 5dpi-Nap: n=2–3 animals; 7dpi-Poli: n=3–4 animals. Student's t-test, **p<0.01, ***p<0.001, ****p<0.0001, n.s., not significant. Scale bars:10µm.

The online version of this article includes the following figure supplement(s) for figure 5:

**Figure supplement 1.** Distinct kinetics of repopulation between the ventral and dorsal BC-derived tracheal epithelium post-Naphthalene (5dpi) and post-Polidocanol (7dpi) injury in adult mice.

**Figure supplement 2.** BC-1 and BC-2 markers maintain their differential ventral-dorsal enrichment in basal cells during repopulation of the airway epithelium post-injury.

**Figure supplement 3.** Regional differences in ventral-dorsal programs of repopulation are no longer evident at late stages of regeneration post injury.

with only 8.69 ± 2.46% ventrally (*Figure 5* and *Figure 5—figure supplement 3C*). Consistent with these observations, quantitative analysis of the perimeter occupied by KRT13 + cells along the basement membrane in the regenerating epithelium was significantly higher dorsally, compared with the ventral trachea (5dpi-Nap: 57.79±5.28% vs 9.25 ± 2.64%; 7dpi-Poli: 71.98±5.13% vs 10.03 ± 3.28%; *Figure 5—figure supplement 3D*). The extensive proliferation and expansion of KRT13 + cells seemed to occur at the costs of differentiation in dorsal BCs.

Interestingly, the differential enrichment of BC-1 and BC-2 markers in ventral and dorsal BCs, respectively, could be clearly distinguished in the epithelium undergoing regeneration in both models (*Figure 5—figure supplement 2*). This further supported the idea that among all BC subtypes, BC-1 and BC-2 are the main populations responsible for epithelial regeneration in response to the severe tracheal injury.

By 15dpi when the epithelium had already extensively regenerated in both models, the proportion of KRT13 +labeled cells was significantly reduced in both the dorsal and ventral compartments (Nap: 5.79±2.19% vs 0.44 ± 0.34%; Poli:10.69±2.53% vs 0.54 ± 0.31%) (Figure S12C). The areas contiguously occupied by KRT13 expressing cells along the basement membrane were also markedly reduced by 15dpi in the dorsal and ventral trachea (Nap: 8.56±3.35% vs 0.72 ± 0.5%; Poli: 10.8±3.01% vs 1.19 ± 0.54%) (*Figure 5—figure supplement 3D*).

Overall, these observations identify a distinct ability of dorsal-derived (BC-2) progenitors to initiate a program of repair in response to environmental perturbations in culture and severe injury in vivo. This program differs from the ventral (BC-1)-associated program as it is characterized by induction of a metaplastic state associated with KRT13 expression and likely to be transient in the absence of persistent injurious stimuli.

## BC heterogeneity is defined during embryonic development

A key observation from our analyses was that the identities and intrinsic behaviors of adult BC-1 and BC-2 cells were maintained when isolated and expanded in culture independent of their niche, for at least 14 days. We, thus, asked whether acquisition of heterogeneity was an early event that paralleled BC fate specification during lung development. We have previously shown that, in the embryonic mouse lung, airway epithelial cells destined to become BCs (pre-BCs) are already specified at an early stage, when airways are still forming, but have to undergo a series of maturation events to become properly functional (*Yang et al., 2018*). Thus, to investigate the developmental basis of BC heterogeneity, we first searched for evidence of differential distribution of the adult BC-1 and BC-2 markers in pre-BCs from ventral and dorsal embryonic tracheas. Tracheas were isolated from E18.5 embryos and triple-labeled for KRT5 with each marker and a-SMA (which served as a reference for the dorsal location). This revealed significant differential enrichment of BC-1 and BC-2 markers, consistent with the pattern identified in the adult trachea. Quantitative analysis of fluorescence intensity, specifically in KRT5-expressing cells, confirmed statistically significant difference of expression for all markers except CAV1 (*Figure 6A and C*).

We then reasoned that local epithelial-mesenchymal crosstalks were crucial in establishing BC heterogeneity as they were being specified and undergoing maturation in the developing trachea. Cartilage primordia appear early in tracheal development (~E10.5) when TRP63 +BC precursors are

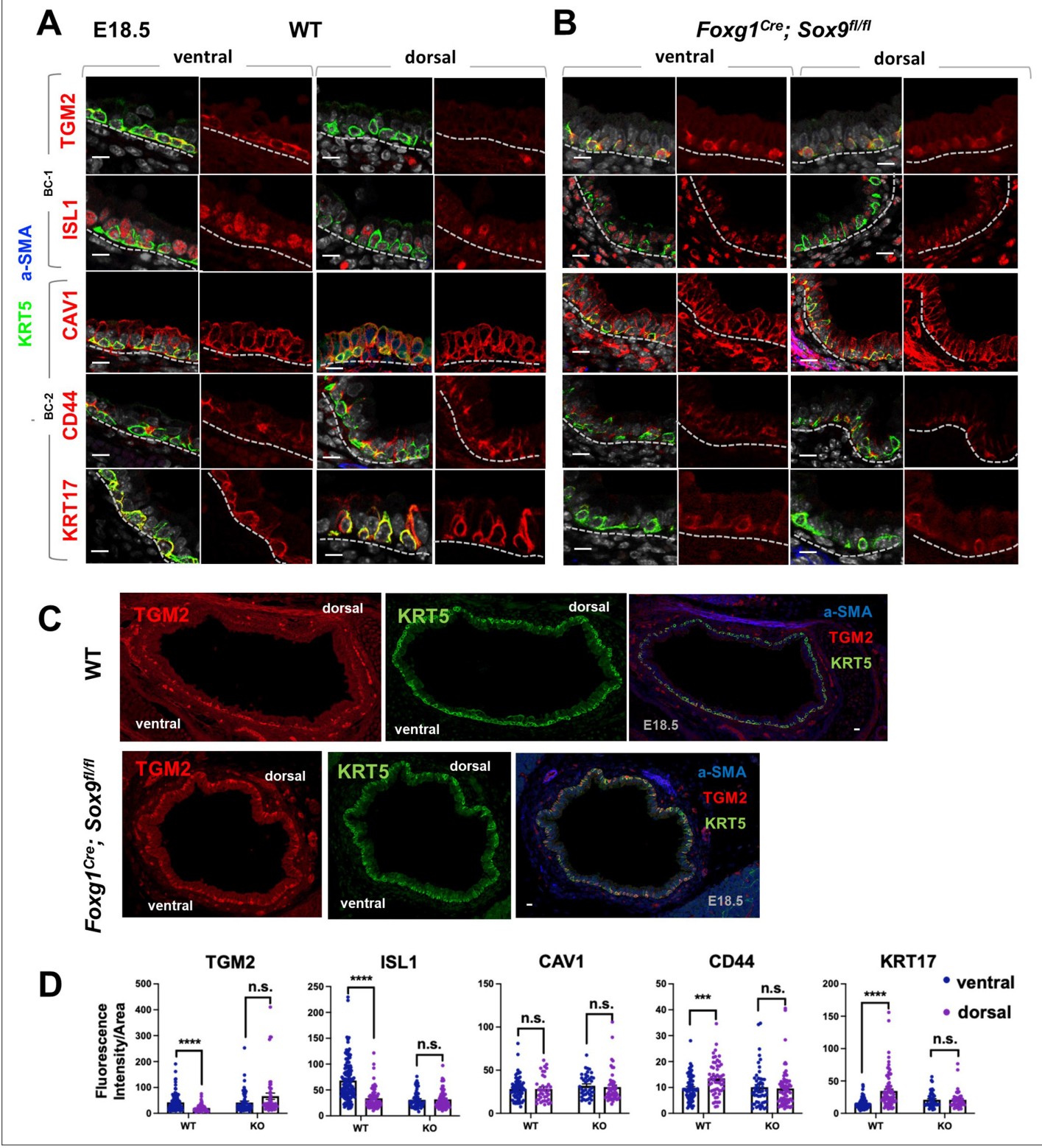

**Figure 6.** Basal cell heterogeneity is established during embryonic development. (**A–B**) IF of BC-1 and BC-2 markers co-labeled with KRT5 in tracheal sections from E18.5 WT and *Foxg1^Cre^; Sox9^fl/fl^* littermates. BC-1 and BC-2 markers maintain the ventral-dorsal differential enrichment of adult WT trachea BCs (a-SMA marks the smooth muscle in dorsal trachea). This pattern is abolished in BCs from E18.5 mutant tracheas. (**C**) Broad view of a cross section from E18.5 WT and mutant tracheas depicting the TGM2 distribution in BCs also seen in (**A**), (**B**). (**D**) Quantification of Fluorescence intensity of markers in E18.5 ventral and dorsal BCs from WT and mutants. Bars represent the mean ± SEM of average fluorescence intensity values in a single KRT5 +BC (dots, already clarified in methods), n=3 animals per genotype. Student's t-test, ***p<0.001, ****p<0.0001; n.s., not significant. Scale bars: 10 μm.

already present but still rather immature, without evidence of additional early markers of BC fate, such as KRT5 or KRT15 expression, or a recognizable BC layer in the developing tracheal epithelium (*Yang et al., 2018*; *Hines et al., 2013*). Genetic studies have demonstrated the crucial role of *Sox9* as a master regulator of cartilage cell fate (*Bi et al., 1999*). Targeted deletion of *Sox9* in the developing tracheal mesenchyme of *Foxg1^Cre; Sox9^fl/fl* mutants results in complete absence of cartilage, a tracheal defect lethal at birth (*Hines et al., 2013*; *Hébert and McConnell, 2000*; *Nasr et al., 2021*; *Bottasso-Arias et al., 2022*). The inability of these mutants to specify cartilage cell fate and to establish a ventral niche while BCs were developing provided a unique opportunity to examine the impact of the local environment in BC-1 vs BC-2 diversification.

Tracheas were isolated from WT and *Foxg1^Cre; Sox9^fl/fl* mutants at E18.5 and analyzed for the dorsal-ventral distribution of the BC-1 and BC-2 markers by IF, as described above. We focused our analysis on E18.5 because of the low levels or inconsistent expression of these markers at earlier stages. As previously reported, tracheal rings or cartilage condensations were absent in *Foxg1^Cre; Sox9^fl/fl* mutants and replaced by an indistinct mesenchymal tissue in the ventral trachea, while, dorsally, a smooth muscle layer was evident (*Hines et al., 2013*; *Hébert and McConnell, 2000*). KRT5 labeling identified pre-BCs in both ventral and dorsal tracheal regions of these mutants, confirming previous reports that disruption of the cartilage niche may lead to an overall decreased number of pre-BCs (*Figure 6A–B*). Tracheal sections triple-labeled with each marker/KRT5/a-SMA showed signals in both ventral and dorsal pre-BC of E18.5 WT and mutants. Remarkably, a comparison of the intensity of signals between ventral and dorsal pre-BCs from mutants showed no significant difference for all markers, in sharp contrast with the findings in WT tracheas (*Figure 6A–C*). We propose this could presumably have resulted from attenuation of local signals normally present in the ventral mesenchyme or cartilage primordia, key in creating the differences in gene expression between BC-1 and BC-2.

The data underscored a key role for epithelial-mesenchymal crosstalks during co-development of the cartilage and the BC pool generating spatial heterogeneity among pre-BCs of the embryonic trachea.

## Airway BC spatial heterogeneity is conserved in human airways

scRNA-Seq analysis of the human lung has also shown that BCs comprise a heterogenous population of multipotent progenitors in the airway epithelium. In spite of some similarities, it was unclear the extent to which mouse and human BCs have common subtypes and share some of the features we identified here. We asked whether the markers of BC-1 and BC-2 that we identified in the murine trachea were also present in distinct BC subpopulations of the human airways.

Thus, we assessed expression of the five representative mouse BC-1 and BC-2 markers in tracheal sections from adult healthy human donors. Double-labeling with KRT5 showed that all but TGM2 were expressed in BCs. Although expressed in human tracheal epithelium, TGM2 was rather present in luminal cells (Data not shown). Quantitation of fluorescence intensity of these signals revealed a pattern of differential enrichment of mouse BC-1 and BC-2 markers similarly present in human ventral and dorsal BCs, respectively (*Figure 7A*). Notably and consistent with our findings in mice, these features continued to distinguish the two BC subpopulations even when they were isolated from their original niches and maintained in culture under the same condition. IF staining of human BCs expanded to confluency showed distinctly stronger ISL1 signals in the cultures isolated from ventral (BC-1) in contrast to the dorsal BC-2-derived cultures, which were readily recognizable by the stronger CAV1, CD44, and KRT17 expression (*Figure 7B*).

Furthermore, when allowed to differentiate under ALI, human dorsal BC-derived cultures consistently generated more KRT13 + cells (*Figure 7C*, 45.59 ± 3.58% compared with 19.67 ± 3.98%). Differences between cultures from these distinct BC subpopulations could be seen even after prolonged time, as shown in ALI day 28 cells. These observations support the idea of potentially conserved mechanisms of BC diversification between mice and humans. A better understanding of these issues could provide information crucial for the development of models relevant to human lung repair-regeneration.

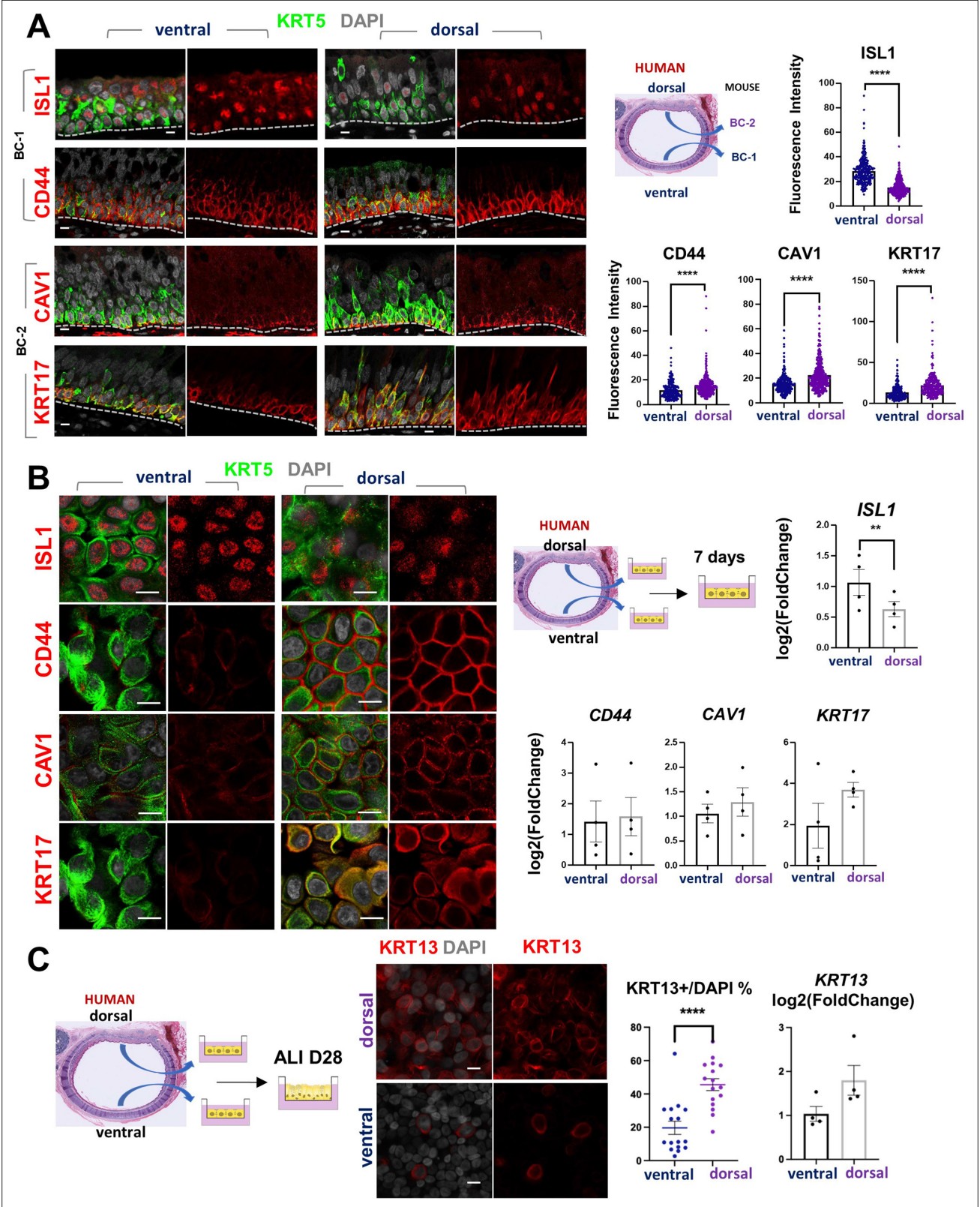

**Figure 7.** Airway basal cell spatial heterogeneity is conserved in human airways. (**A**) IF of ISL1, CD44, CAV1 and KRT17 co-stained with KRT5 in adult human tracheal sections. Quantification of fluorescence intensity in ventral and dorsal BCs from paired regionally distinct areas. Bar graphs are mean ± SEM of the values from each BC (dots), n ≥ 227 BCs from 3 different human donors. (**B**) Differential enrichment of BC-1 and BC-2 markers in ventral and dorsal cultured human BCs, respectively. littermate control of BC-1 and BC-2 markers in 7 days submerged cultures of BCs isolated from ventral

*Figure 7 continued on next page*

*Figure 7 continued*

and dorsal sides of adult human donor tracheal epithelium. qPCR of *ISL1, CD44, CAV1, and KRT17* expression in these cultures. Bar graphs are mean ± SEM log2(foldchange) values for each replicate culture (n=3). Data are normalized to ventral BCs. (**C**) IF of KRT13 in ALI day 28 cultures of human ventral and dorsal tracheal BCs. Scattered plot (left): percentage of KRT13 + cells per field in total ventral or dorsal BC population; mean ± SEM from 5 to 6 fields per cultures, n=3 batches. Bar graph (right): qPCR of *KRT13* expression in ALI day 28 cultures from ventral and dorsal human BCs. Values (log2(foldchange)) are normalized to ventral BCs. Bars are mean ± SEM, dots representing values of each replicate. n=4 batches. Student's t-test, **p<0.01, ****p<0.0001. Scale bars represent 10 µm.

## Discussion

In the present study, we report a broader range of heterogeneity of BCs in the progenitor cell pool of the adult murine airways under homeostasis. We show that in spite of this diversity, most of the BCs comprise two major subpopulations, which are similarly uncommitted, compared with the other BC subpopulations. These uncommitted progenitors exhibit comparable levels of canonical BC markers but have distinct signatures and differ from each other in their ability to initiate a metaplastic program of differentiation. Remarkably, their intrinsic differences in phenotype appear to be largely determined by their spatial localization and established when BCs are still developing in the embryonic trachea. Once established, airway BCs can maintain their subtype-specific features to adulthood and behave distinctly in response to different perturbations in vitro as well as injury challenges to the airway epithelium in vivo.

As adult stem cells, airway BCs continuously self-renew and differentiate into luminal lineages during homeostasis. Rather than characterizing BC diversity based on their differentiation potential predicted by trajectory analysis, here we identified the various subpopulations based on their *bona fide* signatures in an unperturbed state. The use of *Trp63$^{CreERT2}$*; *Rosa26$^{lox-STOP-lox-tdTomato}$* mice in our scRNA-Seq screen allowed the identification of a broader range of BCs compared with similar screens using *Krt5$^{CreER}$* based lines, given previous reports of KRT5 negative cells in the BC pool (*Yang et al., 2018*; *Xi et al., 2017*). Recent scRNA-Seq studies have identified a panel of BC markers that includes *Hlf, Icam1, Notch1, Ngfr, Krt15, F3, Aqp3,* besides *Krt5,* also present in our screen (*Plasschaert et al., 2018*; *Montoro et al., 2018*). While we found these genes widely distributed in all clusters, none of them distinguished the two main BC subpopulations identified in our study (*Figure 1—figure supplement 3*).

Spatial heterogeneity has been reported in human BCs isolated from nasal, trachea, and intrapulmonary airways with region-specific phenotypes shown to be maintained in culture over extended time (*Kumar et al., 2011*). A surprising finding from our report was the unbiased identification of BC-1 and BC-2, the two main BC subpopulations, differentially distributed in the dorsal and ventral trachea, in agreement with the findings from a recent study in which BCs were isolated and analyzed primarily from these regions (*Tadokoro et al., 2021*). Our data support the idea that heterogeneity is assigned while BCs are specified and may involve epigenetic modifications. Detailed characterization of chromatin accessibility, histone modification and DNA methylation may further unveil the molecular mechanisms involved in the generation and maintenance of the BC-1 and BC-2 phenotypes. scRNA-Seq of the corresponding niches should also provide valuable information about intercellular signal communications critical in the establishment of BC heterogeneity. Furthermore, tensile/structural properties (e.g., stiffness, collapsibility) of the membranous (dorsal) trachea vs cartilaginous (ventral) trachea could also contribute to the intrinsic differences in cell behavior observed between BC-1 and BC-2. This would be difficult to test in adult intact animals, as the inability to form tracheal rings (such as in the *Sox9* mutants here) is lethal at birth.

Phenotypic differences have been reported in BCs of the ventral tracheal epithelium immediately adjacent to the cartilage rings compared with those present in the inter-cartilaginous regions in which the mesenchyme exhibits a smooth muscle layer (*Li et al., 2015*; *Volckaert and De Langhe, 2014*; *Volckaert et al., 2017*). While both BCs are present in the ventral epithelium, our markers could not consistently distinguish cartilage-associated from the inter-cartilaginous BC subpopulations. This could be ascribed to the fact that the BC-1 and BC-2 markers identified in our unbiased approach were expressed in both ventral and dorsal subpopulations at different levels, but not enough to consistently distinguish cartilaginous vs non-cartilaginous BCs in our assays. Moreover, these experiments were not designed to isolate and study specifically inter-cartilaginous BCs.

The potential clinical significance of these findings is intriguing as they point to a differential susceptibility of the BC-1 and BC-2-derived epithelia to activate a potentially pathological program associated with *Krt13*. Squamous metaplasia is a common pre-neoplastic finding in smokers and patients with chronic obstructive pulmonary diseases (COPD). Although reversible, under continued injury the metaplastic epithelium can undergo malignant transformation to lung squamous cell carcinoma. Interestingly, several genes found to be differentially enriched in the BC-2 subpopulation, such as *Tppp3*, *Tnfrsf12a*, and *Cav1* have been associated with severity or aggressiveness of squamous carcinoma in various organs (*Yang et al., 2020*; *Hu et al., 2019*; *Xue et al., 2010*). Additionally, a recent study identifies metaplastic variant TRP63-expressing clones in COPD patients and in fetal human lungs (*Rao et al., 2020*).

Future studies exploring the impact of BC heterogeneity in the molecular mechanisms of initiation of tumorigenesis can provide significant insights into the pathogenesis of human conditions.

## Methods

### Mouse genetic models

*Trp63^{CreERT2}*; *Rosa26^{lox-STOP-lox-tdTomato}* mice were generated and characterized as described (*Yang et al., 2018*; *Lee et al., 2014*). To label airway BCs at homeostasis, 6–12 week-old mice were exposed to 240 μg/g body weight TM once via oral gavage. A short chase period of 3 days was used to ensure accumulation of tdTomato in BCs and to minimize tdTomato labeling in luminal descendants.

To generate *Foxg1^{Cre}*; *Sox9^{fl/fl}* embryos, *Sox9^{fl/fl}* mice (*Kist et al., 2002*) were mated with *Foxg1^{Cre}* mice (*Hébert and McConnell, 2000*) to generate *Foxg1^{Cre}*; *Sox9^{fl/fl}* embryos as previously described *Nasr et al., 2021*; *Bottasso-Arias et al., 2022*. Genotypes of transgenic mice were determined by PCR using genomic DNA isolated from mouse-tails or embryonic tissue (*Sox9* F: 5' CCG GCT GCT G GG AAA GTA TAT G 3'. *Sox9* R: 5' CGC TGG TAT TCA GGG AGG TAC A 3'. *Foxg1-Cre* F: 5' TGC C AC GAC CAA GTG ACA GCA ATG 3'. *Foxg1-Cre* R: 5' AGA GAC GGA AAT CCA TCG CTC G 3'). For embryonic developmental staging, the morning when vaginal plug was detected was considered E 0.5. Animals were housed in a pathogen-free environment and handled according to the protocols approved by CCHMC Institutional Animal Care and Use Committee (Cincinnati, OH, USA).

All studies were approved by the Columbia University Institutional Animal Care and Use committees (WVC IACUC #: AC-AABF2567,) and CCHMC Institutional Animal Care and Use Committee (DS IACUC #: 2021–0053).

### Basal cell isolation and fluorescence activated cell sorting (FACS)

Tracheas were dissected from adult *Trp63^{CreERT2}*; *Rosa26^{lox-STOP-lox-tdTomato}* mice after 72 h of TM administration. For BC isolation, only the region below the cricoid cartilage and above the carina were used to avoid including TRP63^+ BCs from the larynx or mainstem bronchi. Tracheal tubes were cut open and digested in Dispase Digestion Solution (16 U/ml Dispase (Corning, 354235)+10 μg/ml DNase I diluted in PBS) at room temperature for 40 min. Digestion was stopped by transferring tracheas to PBS. Epithelium was physically peeled off with forceps, and collected in Falcon tubes. Further digestion into single cells was done in Trypsin Digestion Solution (0.1% Trypsin +3 mM EDTA diluted in PBS) at 37 °C for 10 min. To stain for cell sorting, cells were suspended in staining buffer (FACS buffer +10 μg/ml DNase I), and incubated with antibodies for 40 min at 4 °C. Cells were washed with FACS buffer, and DAPI was added to a final concentration of 1.25 μg/ml before sorting. Sorting was performed on Influx (BD Biosciences) and data analyzed with Flowjo (version 10). Cells were collected in staining buffer. The following antibodies were used: CD104-BV510 (1:50, BD Biosciences, 743079), EPCAM-APC (1:100, eBioscience, 17-5791-82), Lineage cocktail-Alexa Fluor 700 (1:20, Biolegend, 133313).

### Single cell RNA sequencing

Immediately after sorting, cells were stored on ice and processed at the Columbia University Genome Center for 10× Chromium single cell sample preparation. Single Cell 3' libraries were prepared using the Chromium Single Cell 3' v2 Protocol (CG00052) according to the manufacturer's manual (10× Genomics). The pooled, 3'- end libraries were sequenced using Illumina HiSeq4000. Cell Ranger version 2.1.1 was used for primary data analysis, including demultiplexing, alignment, mapping, gene

expression quantification, dimension reduction analysis, and clustering analysis within individual data-sets. Specifically, for alignment and mapping, the mm10 reference genome and corresponding annotation were used.

## Gene set overlapping analysis and GSEA

BC-1 and BC-2 specific differentially expressed genes highlighted in *Figure 1F* (blue and orange, respectively) were used for the analysis. The analysis was performed using Broad Institute web-based gene set overlapping analysis tool, as available at http://software.broadinstitute.org/gsea/msigdb/annotate.jsp. KEGG gene sets were included in the analysis, and FDR q-value threshold was set as 0.05.

The squamous metaplasia signature gene set was generated by converting human smoking-associated squamous metaplasia genes (*Goldfarbmuren et al., 2020*) to mouse homologs. BC-1 and BC-2 gene expression values from these two clusters from scRNA-Seq profiles were used for comparison. GSEA was performed following the developer's protocol (GSEA version 4.1.0, available at: http://www.gsea-msigdb.org/gsea/index.jsp).

## Immunofluorescence staining and confocal analysis

Embryonic and adult tracheas were fixed in 4% paraformaldehyde in PBS at 4 °C overnight. After washing in PBS, samples were processed for frozen or paraffin-embedding.

Immunofluorescence (IF) was performed in tissue sections (6–8 µm) blocked with 1% bovine serum albumin (Sigma) and 0.5% TritonX-100 (Sigma) for 1 hr at room temperature. Primary antibodies were incubated in 1% bovine serum albumin (Sigma) and 0.5% TritonX-100 at 4 °C overnight. Sections were then washed with PBS and incubated with Alexa Fluor-conjugated secondary antibodies (1:300) and NucBlue Live Cell ReadyProbes Reagent (DAPI) (Life Technology) for 1 hr. After washing, samples were mounted with ProLong Gold antifade reagent (Life Technology). When necessary, antigen unmasking was done using Citric Based solution (Vector Labs H-3300) heated in microwave. Mouse primary antibody staining was done using M.O.M kit (Vector Labs BMK-2202).

The following primary antibodies were used: rabbit anti-TRP63a (1:100, CST, 13,109 s); chicken anti-KRT5 (1:300, Biolegend, 905901); rabbit anti-KRT17 (1:10000, Abcam, ab53707); rabbit anti-CAV1 (1:100, Cell Signaling Technology, 3267 S); rat anti-CD44 (1:100, BD Biosciences, BD553131); rabbit anti-TGM2 (1:50, Cell Signaling Technology, 3557 S); rabbit anti-ISL1 (1:50, Abcam, ab109517); rat anti-SCGB3A2 (1:100, R&D Systems, MAB3465); goat anti-CC10 (1:1000, Santa Cruz biotechnology, sc-9772); mouse anti-ECAD (1:200, BD Biosciences, 610181); mouse anti-FOXJ1 (1:100, eBioscience, 14-9965-82); rabbit anti-Acetyl-α-TUBULIN (1:2000, Cell Signaling Technology, 5335 S); rabbit anti-VIMENTIN (1:200; Cell Signaling Technology; 5741); rat anti-KI67 (1:100, eBioscience, 14-5698-82).

The following secondary antibodies were used: donkey anti-rabbit (conjugated with Alexa Fluor 488, 568, 647); donkey anti-chicken (conjugated with Alexa Fluor 488); goat anti-chicken (conjugated with Alexa Fluor 488, 647); donkey anti-mouse (conjugated with Alexa Fluor 488, 568, 647); donkey anti-rat (conjugated with Alexa Fluor 488, 647); donkey anti-goat (conjugated with Alexa Fluor 488, 647). All secondary antibodies were purchased from Thermo Fisher Scientific or Jackson ImmunoReseach.

Mouse Air-liquid interface culture inserts were fixed in 4% paraformaldehyde in PBS at room temperature for 15 min (day 0) or 30 min (day 7). The insert membranes were cut into 6–8 pieces, and subsequently blocked and stained with antibodies as described above.

Human Air-liquid interface culture inserts were fixed in 4% paraformaldehyde in PBS at room temperature for 1 hr (day 0) or overnight (day 28). The insert membranes were cut into 6–8 pieces, processed for antigen unmasking using Citric Based solution (Vector Labs H-3300) when required, subsequently blocked and stained with antibodies as described above.

Organoid cultures (tracheospheres) were fixed in 4% paraformaldehyde in PBS at room temperature for 30 min and then processed for frozen sections. Organoid sections (9 µm) were blocked and stained with antibodies as described above. Confocal microscopy was performed using a Zeiss LSM 710 confocal microscope through 20×, 40× , or 63× lens.

## β-galactosidase staining of frozen sections

RARE-LacZ transgenic mice were previously described *Rossant et al., 1991*. Adult tracheas were fixed in 4% paraformaldehyde in PBS at 4 °C overnight. After washing in PBS, samples were embedded in OCT and processed for frozen sections (9 µm). The slides were washed in 0.1% Tween in PBS for 20 min, then the β-galactosidase staining was performed overnight with X-gal substrate. Subsequently, some slides were processed for immunostaining.

## Quantification of fluorescence Intensity in immunostained sections

To compare the intensity of fluorescence signals for each marker in tracheal BCs located on ventral side (associated with cartilage rings) to those on dorsal side (associated with smooth muscle layer), sections were immunostained with KRT5 and the specific marker. Using confocal microscopy, paired images with the same z axis were acquired in the same section using the same digital settings. For each set of paired views, 10–20 BCs were randomly picked, circled around the KRT5 labeled cellular shape. Background fluorescence intensity was determined for each image by picking areas without cells or any noticeable fluorescence signals. Fluorescence intensity and area ($mm^2$) for each cell were measured using Zen 2.3 lite software. The intensity value of each marker channel per unit area for each BC was calculated and analyzed for statistical significance For each marker, 8–15 fields from 5 to 9 sections from 3 different animals were analyzed. To compare the fluorescence intensities of each marker in the mutant (*Foxg1$^{Cre}$; Sox9$^{fl/fl}$*) and control mice at developmental E18.5, cross-sectional immunofluorescence staining images were acquired using confocal microscopy. The average fluorescence intensity was measured as described above.

## Isolation and analysis of tracheal BCs in ALI organotypic cultures

Tracheas from adult mice or from adult human organ donors were resected, and the membranous (dorsal) and cartilaginous (ventral) regions were surgically separated under a dissecting microscope. BC populations from each of these regions were isolated using established protocols after Pronase (1.5 mg/mL) digestion and differential adhesion, and cultured separately.

For the mouse ALI cultures, equal amounts of basal cells ($3.3x10^4$ cells) from each region were seeded in Transwell plates and expanded initially under submerged conditions in Mouse Tracheal Epithelial Cell (MTEC) Plus medium for 7 days (day –7 to day 0) and then subsequently cultured under ALI conditions in MTEC serum- free media for additional 7 days (ALI day 0 to ALI day 7). To compare the effect of disrupting RA signaling in BC-1 vs BC-2, cells from ventral (BC-1) and dorsal (BC-2) regions were cultured in the presence of the pan RAR reverse agonist BMS493 (1 uM) from day –7 to day 7. Controls were treated with DMSO. Cultures were harvested at ALI day 7 and processed for qPCR (below) and IF. We performed immunostaining for KRT13 in these cultures and counted the number of KRT13 + cells per field in more than 12 fields per experimental condition (BC-1 vs BC-2 in control and BMS-treated cultures) in three different experiments. The percentage of KRT13 labeling was calculated as the number of KRT13 +per total number of DAPI positive cells.

For the human BC cultures, equal amounts of tracheal BCs ($3.3x10^4$ cells) from the dorsal and ventral regions of an adult human trachea were isolated similarly as above and cultured under submerged conditions in Bronchial epithelial growth medium (BEGM) for 7 days. For some experiments, confluent BC cultures (ventral and dorsal) were harvested and analyzed for markers of expression of BC-1 and BC-2 markers by qPCR. For other experiments, dorsal-derived and ventral-derived BC cultures were allowed to differentiate under ALI conditions in BEGM-based medium and harvested at ALI day 28. Cultures were processed for KRT13 immunostaining and quantitative analysis and for qPCR analysis of *Krt13*.

## Generation and analysis of mouse 3D organoid cultures

Tracheal BCs were isolated from surgically-dissected dorsal and ventral locations of adult wild-type C57BL/6 mice as described above. Then, $3x10^3$ cells from each region were resuspended in MTEC/Plus medium, mixed 1:1 with growth factor-reduced Matrigel, and seeded into Transwell inserts. MTEC/Plus was added to the lower chamber and BCs were cultured for 7 days; then, the media was replaced by MTEC/SF, and cultures were harvested after another 7 days as previously reported *Rock et al., 2009*. Tracheospheres were harvested at day 14 and processed for IF (KRT5, KRT13, VIM). Analysis of VIM staining showed no evidence of mesenchymal contamination (Figure S8).

To quantify organoid colony forming efficiency, the total number of tracheospheres per well in BC-1 (ventral) or BC2 (dorsal)-derived cultures were counted in 6 independent cultures from each group using an EVOS m5000 microscope (Invitrogen) at 4× magnification. The organoid forming efficiency (%)=total organoid number / total (3000) seeding cells was determined for BC-1 and BC-2. For the quantification of organoid size, the diameter of all organoids in a random field per well was measured in BC-1or BC-2-derived tracheospheres using Fuji ImageJ (*Zhou et al., 2018*). The individual size of 10–77 organoids was measured in each dish and the average size of BC-1 cultures in 6 batches (Mean ± SEM: 35.92±1.00, n=236 organoids) was compared to BC-2 cultures (Mean ± SEM: 35.37±1.15, n=268 organoids). The average size of all organoids (35.65) was used as a cutoff to determine whether an organoid was classified as small (<35.65) or large (>35.65). The percentages of small or large organoids for each group from each batch were then determined as the number of small or large-size organoids divided by the total number of organoids in that specific batch for each group. The amount KRT13 +or KRT5 +labeling was quantified in day 14 tracheospheres derived from ventral (BC-1) or dorsal (BC-2) cells and expressed as percentage relative to all DAPI + cells using confocal microscopy and the Zen 2.3 lite software.

## q-PCR

RNA was extracted using QIAGEN RNeasy Mini Kit and cDNA was synthesized using the SuperScript IV First-Strand synthesis system (Thermo Fisher). The following primers (Thermo Fisher) were used: mouse *Krt13* (Mm00495199), mouse *Tgm2* (Mm00436987_m1), mouse *Krt17* (Mm00495207_m1), mouse *Cav1* (Mm00483057_m1), mouse *Cd44* (Mm01277163_m1), mouse *Islet1* (Mm00517585_m1), mouse *Col17a1*(Mm00483525_m1), mouse *Sprr1a* (Mm01962902_s1), mouse *Loricrin* (Mm01962650_s1), human *KRT13* (Hs00357961_g1), human *ISLET1* (Hs00158126_m1), human *KRT17* (Hs00356958_m1), human *CAV1* (Hs00971716_m1), and human *CD44* (Hs01075864_m1). Reactions were performed using Taq-Man Advanced Master Mix (Thermo Fisher #4444556) using *b-Actin* (for mouse samples) and *GAPDH* (for human samples) as internal control and a Step-One Plus Instrument (Applied Biosystems). DDCT method was used to calculate changes in expression levels. At least three biological repeats were analyzed for each group.

## Naphthalene/polidocanol injury mouse models

Wild-type C57BL/6 6–12 week-old mice were used for induction of naphthalene or polidocanol injury -repair. Only females were used for naphthalene injury. Both male and females were used for polidocanol injury. The studies were approved by Columbia University Institutional Animal Care and Use committees (IACUC).

For Naphthalene injury, a single dose of naphthalene (Nap, dissolved in sunflower oil) was administered to female wild-type mice by IP injection at 275 mg/kg body weight to induce trachea injury. Freshly prepared naphthalene was administered before noon (*Tata et al., 2018*). For Polidocanol injury, wild-type mice were anesthetized and received 20 ul of 2% polidocanol (poli, freshly prepared in PBS) by oropharyngeal aspiration delivery following previously published protocols (*Paul et al., 2014*). Animals were sacrificed at day 5 and day 15 (Nap), or day 7 and day 15 (Poli) post injury. Paired images from both dorsal and ventral sides with the same z-axis were acquired in the same section using the same digital setting to quantify the percentages of KRT13+, KRT5+, CC10+, SCGB3A2+, and FOXJ1 + cells. Paired images were also used in quantification of proliferative index of TRP63 +BCs with anti-KI67 antibody.

## Quantification and statistical analyses

Quantification and statistical analysis have already been detailed in the Methods section above and in the figure legends. All quantification for colocalization and marker analyses were performed in Adobe Photoshop. Statistical analyses were performed in Microsoft Excel or GraphPad Prism.

## Acknowledgements

We would like to thank all members of Cardoso lab and the members of the CCHD, Jianwen Que, Munemasa Mori and Hans Snoeck for thoughtful discussions. We also thank Erin Bush from the Columbia Genome Center for help in the single cell RNA-Seq experiment. This work was supported by NIH-NHLBI R35-HL135834-01 to WVC, NIH-NHLBI RO1-144744 to DS, NIH S10OD020056 to

CCTI (Columbia Center for Translational Immunology) Flow Cytometry Core and NIH/NCI Cancer Center Support Grant P30CA013696 to HICCC (Herbert Irving Comprehensive Cancer Center), and the National Center for Advancing Translational Sciences NIH-UL1TR001873 to CTSA Translational Science Award.

## Additional information

### Funding

| Funder | Grant reference number | Author |
| --- | --- | --- |
| National Institutes of Health | R35-HL135834-01 | Wellington V Cardoso |
| National Institutes of Health | RO1-144744 | Debora Sinner |
| Flow Cytometry Core and NIH/NCI Cancer Center | P30CA013696 | Andrea Califano |
| National Center for Advancing Translational Sciences | UL1TR001873 | Andrea Califano |
| National Institutes of Health | R35 CA197745 | Andrea Califano |
| National Institutes of Health | S10 OD012351 | Andrea Califano |
| National Institutes of Health | S10 OD021764 | Andrea Califano |

The funders had no role in study design, data collection and interpretation, or the decision to submit the work for publication.

### Author contributions

Yizhuo Zhou, Conceptualization, Formal analysis, Investigation, Writing – review and editing; Ying Yang, Conceptualization, Formal analysis, Investigation, Writing – original draft, Writing – review and editing; Lihao Guo, Software, Investigation, Visualization; Jun Qian, Validation, Investigation; Jian Ge, Investigation; Debora Sinner, Resources; Hongxu Ding, Supervision, Investigation, Visualization, Writing – original draft, Writing – review and editing; Andrea Califano, Supervision, Writing – review and editing; Wellington V Cardoso, Conceptualization, Supervision, Funding acquisition, Investigation, Writing – original draft, Writing – review and editing

### Author ORCIDs

Ying Yang (ID) http://orcid.org/0000-0002-4197-6216
Debora Sinner (ID) http://orcid.org/0000-0002-0704-5223
Hongxu Ding (ID) http://orcid.org/0000-0002-7846-9744
Andrea Califano (ID) http://orcid.org/0000-0003-4742-3679
Wellington V Cardoso (ID) http://orcid.org/0000-0002-8868-9716

### Ethics

All studies were approved by the Columbia University Institutional Animal Care and Use committees (WVC IACUC #: AC-AABF2567,) and CCHMC Institutional Animal Care and Use Committee (DS IACUC #: 2021-0053).

### Decision letter and Author response

Decision letter https://doi.org/10.7554/eLife.80083.sa1
Author response https://doi.org/10.7554/eLife.80083.sa2

## Additional files

### Supplementary files
• MDAR checklist

### Data availability
scRNA-Seq data for mouse trachea BCs described in the manuscript have been deposited at the Gene Expression Omnibus (GEO) under accession number GSE134064. It can also be explored through MmTrBC data portal at http://visualify.pharmacy.arizona.edu/MmTrBC/.

The following dataset was generated:

| Author(s) | Year | Dataset title | Dataset URL | Database and Identifier |
| --- | --- | --- | --- | --- |
| Yang Y, Cardoso W, Ding H, Califano A | 2019 | scRNA-Seq of Mouse Trachea Basal Cells | https://www.ncbi.nlm.nih.gov/geo/query/acc.cgi?acc=GSE134064 | NCBI Gene Expression Omnibus, GSE134064 |

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
