## [Editor Report]

This study provides key evidence for basal cell heterogeneity in the airways of the lung. Critical evidence of transcriptional and signaling pathways that regulate the spatial patterning of ball cell heterogeneity are revealed. This information will hopefully guide a new understanding of how basal cells in the lung airways drive metaplastic diseases including cancers.

---

## [Decision Letter]

**Decision letter after peer review:**

Thank you for submitting your article "Airway Basal Cells Show Regionally Distinct Potential to Undergo Metaplastic Differentiation" for consideration by *eLife*. Your article has been reviewed by 3 peer reviewers, and the evaluation has been overseen by a Reviewing Editor and Edward Morrisey as the Senior Editor. The following individuals involved in the review of your submission have agreed to reveal their identity: Jason R Spence (Reviewer #1); Darrell N Kotton (Reviewer #2).

Essential revisions:

1) Please add the additional expression data as requested by the reviewers i.e. immunostaining and QPCR were indicated.

2) Please edit the manuscript to add additional clarification of data and methods where requested.

*Reviewer #1 (Recommendations for the authors):*

– Were there any markers that were expressed in a binary on/off fashion that can be used to distinguish BC1 vs. BC2. Currently, the data supports that mRNA or protein changes between differentially expressed markers are significant but subtle. If there were more clear binary markers, it would be useful for the field to more easily identify specific subpopulations.

– Can the authors show the dorsal and ventral nature of staining using two or more (I.e. TGM2, CAV1) markers on a transverse section of the trachea? Is there a sharp boundary for these markers at the mid-line?

– What measures were taken to ensure the purity of BC1 and BC2 of the surgically dissected dorsal and ventral trachea (i.e. for Figure 2). The data shown in Figure 2B is after 7 days in culture, but was any characterization done immediately after sorting to show enrichment of these populations?

– For Figure 2G, the assumption from reading the text is that the same number of starting cells was used for tracheosphere initiation; however, this is not specifically stated in the text. Can the authors please clarify?

– The authors refer to the KRT13 expression in BC2 organoids (Figure 2K) as an "aberrant differentiation program"; however, this reviewer wonders if this is actually the normal differentiation program for BC2s, and how the author can distinguish a normal from abnormal differentiation program in these cells? The data presented in Figure 3 (injury models) show KRT13 activation followed by resolution at day 15. This may indicate that an increase in KRT13 followed by a resolution in BC2 (dorsal) tracheal basal cells is part of the natural injury repair process and not an aberrant program.

– The LOF experiments in Figure 4 are striking. Can the authors show supporting evidence using another quantitative approach such as qRT-PCR?

– Can primitive basal cells (I.e. early embryonic) be patterned in vitro by placing them in co-culture with cartilage or smooth muscle?

*Reviewer #2 (Recommendations for the authors):*

Some caveats for the authors to consider to make this work even stronger are:

1. A panel of useful markers is established in figure 1 that potentially distinguishes BC-1 from BC-2 (e.g. Cd44, Cav1, Krt17, Isl1, Tgm2) – however, these markers are not consistently used throughout the manuscript despite their emphasis early in the story. Could the authors provide an additional mention and comparison more consistently throughout? For example, figure 1D would be much improved by adding this panel of markers to the profiles. The injury response studies (figure 3) in vivo would also benefit from a reprise of the kinetics of expression of these markers post-injury.

2. Beyond the molecular profiles, some of the functional differences seem mild/subtle, even if statistically significant (e.g. CFE differences between BC-1 and BC-2 during culture outgrowth are scored as 7% and 10%, a subtle difference, with no difference in organoid sizes after outgrowth). What do these subtle differences mean and can the authors comment on their relevance to in vivo proliferative capacity – e.g. is there any difference in terms of proliferative responses in vivo after injury that authors could quantify, e.g by regional Edu uptake? In addition, it is unclear to this reviewer what quantitative differences in differentiation repertoires there might be between BC-1 and BC-2. Krt13 is one of the significant differences between BC-1 and BC-2, but how about the expression patterns of other cell lineage makers after ALI culture?

3. In some places the terminology chosen by the authors is confusing or unexplained. For example, the "signatures" of BC-1 and BC-2 "appeared to be the most stem-like BC populations" – it is not clear what the authors mean by "stem-like" and whether they are referring to specific markers (not mentioned) or functionality (not explicitly tested for stemness.) Consider explaining or deleting.

4. The authors emphasize that dorsal and ventral basal cells are different because of different "niches", without studying what these niches or signals might be. Have the authors considered that the different structures, tensile/structural properties (e.g. stiffness, collapsibility) of the membranous (dorsal) trachea vs cartilaginous (ventral) trachea may contribute to epithelial intrinsic differences in cell behavior without differential niche signals. The removal of cartilage (*Sox9* deletion studies in vivo) would then remove these structural differences by changing tensile properties of the local tissue (rather than removing a signaling niche), thus minimizing BC differences, as the authors observed in their knockout tracheas. Could the authors discuss or test for these alternative possibilities since conventional niche-stem cell signaling differences (e.g. "cartilage-derived signals") were not formally assayed in the present work to warrant such strong conclusions about distinct biological niches.

5. How different is the RA signaling activation between BC-1 vs. BC-2 or ventral vs. Dorsal regions? The authors demonstrated that RA signaling plays important roles in Krt13 expression by adding RA inhibitor, BMS, into ALI culture media, suggesting the importance of the niche environment for the determination of the cellular identities, BC-1 and BC-2. However, it is still unclear whether RA signaling is highly activated in BC-1 compared to BC-2. In addition, which niche cells such as chondrocytes or fibroblast secrete RA to BC-1 is unclear.

*Reviewer #3 (Recommendations for the authors):*

The manuscript would benefit from some additional information/experiments.

1) Material and methods seem to indicate that the basal cells isolated for organoid and ALI cultures were not sorted. Figure 2H shows some contamination with stromal cells, especially for the BC2 cultures. It is, therefore, possible that signals from these stromal cells are maintaining these BC1 vs BC2 phenotypes. To address this the authors may need to co-culture GFP positive BC1 isolated from a mouse in which all cells or basal cells are GFP + with Tomato positive BC2 isolated from a mouse in which all cells or basal cells are tomato +. Alternatively, basal cells could be isolated from mice in which Foxg1 positive cells express the Diphtheria toxin receptor so that stromal cells can be killed in culture.

2) What are the different signals coming from the dorsal vs ventral stromal niches? Can the authors give some examples of differentially expressed genes for the different pathways described in Figure 1F?

3) Material and methods also describe the isolation of esophageal basal cells but this experiment does not seem to be included in this manuscript.

4) Where are the other 4 basal cell populations located?

5) This sentence "By contrast, BCs of the dorsal trachea were marked by strong expression of Cd44, Cav1, Krt17, which were only weakly expressed in ventral BCs (Figure 1G)." needs to be rephrased.

---

## [Author Response]

Reviewer #1 (Recommendations for the authors):– Were there any markers that were expressed in a binary on/off fashion that can be used to distinguish BC1 vs. BC2. Currently, the data supports that mRNA or protein changes between differentially expressed markers are significant but subtle. If there were more clear binary markers, it would be useful for the field to more easily identify specific subpopulations.

Our analyses consistently revealed differential enrichment of these markers in the trachea both prenatally and in the adult, but none was expressed in a clear binary (on/off) fashion.

– Can the authors show the dorsal and ventral nature of staining using two or more (I.e. TGM2, CAV1) markers on a transverse section of the trachea? Is there a sharp boundary for these markers at the mid-line?

As requested by the reviewer, we now include new data showing immunofluorescence staining for all markers in cross-sections of the adult trachea (Figure 2—figure supplement 1). The D-V differential enrichment is much clear in these preparations. We can identify defined boundaries of staining in some cases. Although these boundaries are consistently associated with the dorsal (smooth muscle layer) tracheal mesenchyme, it is overall difficult to define exactly where they occur. We ascribe this to differences in the orientation of the histological sections which could not be precisely positioned for this purpose.

– What measures were taken to ensure the purity of BC1 and BC2 of the surgically dissected dorsal and ventral trachea (i.e. for Figure 2). The data shown in Figure 2B is after 7 days in culture, but was any characterization done immediately after sorting to show enrichment of these populations?

We were unable to perform any characterization of these cells prior to their expansion in culture due to the minute amounts of freshly-isolated BCs obtained from surgically-dissected tissue samples. This was particularly challenging for the dorsal BCs, given the much smaller membranous region of the trachea. Thus, we relied in our highly efficient protocols of tissue microdissection, differential adhesion and expansion of these BCs for obtaining pure populations of these cells. The efficiency of our protocols was consistent with our findings of differential behavior between these two BC populations both in 3D organoid assays and in air-liquid interface organotypic cultures. Notably, we now provide evidence that these organoids are not contaminated with mesenchymal cells (Figure 3—figure supplement 1).

– For Figure 2G, the assumption from reading the text is that the same number of starting cells was used for tracheosphere initiation; however, this is not specifically stated in the text. Can the authors please clarify?

This is correct. Information is now stated in the revised manuscript in the main text as well as the methods section.

– The authors refer to the KRT13 expression in BC2 organoids (Figure 2K) as an "aberrant differentiation program"; however, this reviewer wonders if this is actually the normal differentiation program for BC2s, and how the author can distinguish a normal from abnormal differentiation program in these cells? The data presented in Figure 3 (injury models) show KRT13 activation followed by resolution at day 15. This may indicate that an increase in KRT13 followed by a resolution in BC2 (dorsal) tracheal basal cells is part of the natural injury repair process and not an aberrant program.

We agree and have revised the text providing additional explanations about its transient appearance as part a program of epithelial regeneration and a program previously associated with squamous differentiation.

– The LOF experiments in Figure 4 are striking. Can the authors show supporting evidence using another quantitative approach such as qRT-PCR?

We found that qRT-PCR analysis of tissue lysates could not reliably assess the quantitative differences in basal cell expression of these markers for the following reasons. As clearly shown in our immunofluorescence stainings, nearly all of our BC-1 and BC-2 markers were also expressed in other epithelial cell populations (parabasal, luminal cells) and/or mesenchymal cells of the developing trachea. The expression of these markers in other cell populations did not necessarily follow the same behavior of that in the basal cells, making comparisons between WT and mutants inaccurate when assessed in cell lysates. Since the differences in expression were remarkable and demonstrated accurately by immunofluorescence, we measured the local levels of fluorescence staining as our quantitative approach.

– Can primitive basal cells (I.e. early embryonic) be patterned in vitro by placing them in co-culture with cartilage or smooth muscle?

We attempted to address this question in embryonic tracheal explant cultures but found complexities that required experiments beyond the scope of our present work. Some of these complexities included: (1) We could not accurately determine when the differences in D-V distribution of these markers were established in the embryonic trachea and whether these differences were triggered by early ventral signals associated with the initiation of the cartilage program or later by inductive signals arising from the cartilage primordia. (2) Our analysis of the *Foxg1Cre*; Sox9f/f did not allow to properly address this question since *Sox9* was deleted constitutively from the earliest Foxg1-expressing mesenchymal cells in the developing trachea. Thus, co-culturing the early embryonic tracheal epithelium with an already formed cartilage was unlikely to result in the expected patterning effects. (3) Further studies are required to determine precisely when the developing pre-basal cell precursors are patterned and diversified in the murine trachea. As we previously reported (Yang et al., Dev Cell 2018), p63+ Nkx2-1+ basal cell precursors can be identified as early as the stage of lung specification in the foregut. Although subsequently restricted to the trachea, these p63+ cells are scattered throughout the pseudostratified epithelium and only later (after E14.5-15.5) are segregated into a layer of prebasal cells. Thus, at the early stages of tracheal development these layers (luminal, basal) are undefined, initially unseparated from the dorsal (esophageal) epithelium and difficult to manipulate.

Reviewer #2 (Recommendations for the authors):Some caveats for the authors to consider to make this work even stronger are:1. A panel of useful markers is established in figure 1 that potentially distinguishes BC-1 from BC-2 (e.g. Cd44, Cav1, Krt17, Isl1, Tgm2) – however, these markers are not consistently used throughout the manuscript despite their emphasis early in the story. Could the authors provide an additional mention and comparison more consistently throughout? For example, figure 1D would be much improved by adding this panel of markers to the profiles. The injury response studies (figure 3) in vivo would also benefit from a reprise of the kinetics of expression of these markers post-injury.

The currently revised Figure 1F incorporates the BC-1 and BC-2 markers recommended by the reviewer. Furthermore, we have performed immunofluorescence analysis of these markers both in the Naphthalene and Polidocanol mouse injury models in vivo; these results are now shown in the revised Figure 5—figure supplement 2.

2. Beyond the molecular profiles, some of the functional differences seem mild/subtle, even if statistically significant (e.g. CFE differences between BC-1 and BC-2 during culture outgrowth are scored as 7% and 10%, a subtle difference, with no difference in organoid sizes after outgrowth). What do these subtle differences mean and can the authors comment on their relevance to in vivo proliferative capacity – e.g. is there any difference in terms of proliferative responses in vivo after injury that authors could quantify, e.g by regional Edu uptake? In addition, it is unclear to this reviewer what quantitative differences in differentiation repertoires there might be between BC-1 and BC-2. Krt13 is one of the significant differences between BC-1 and BC-2, but how about the expression patterns of other cell lineage makers after ALI culture?

Our observation of a higher colony-forming efficiency in dorsal (BC-2)-derived organoids is in agreement with data from a previous study that showed increased colony-forming efficiency in BCs isolated from the dorsal trachea compared to ventral BCs in similar organoid assays (Tadokoro et al., 2021).

As requested by the reviewer, we have examined the proliferative capacity of these populations in vivo in models of injury-repair. We now show that the dorsal (BC-2) population has a higher proliferative capacity (shown by Ki67 labeling) during initiation of repair and that is particularly striking after Polidocanol injury (Figure 5 and Figure 5—figure supplement 1). Furthermore, we have also investigated the quantitative differences in the differentiation repertoires of BC-1 and BC-2 in these injury models. We show that the increase in the dorsal Krt13 population seen post-Polidocanol (and to a lesser extent post-Naphthalene) is accompanied by a major decrease in number of secretory (CC10, Scgb3a2) and multiciliated (Foxj1) cells (Figure 5—figure supplement 1).

Lastly, as also requested by the reviewer, we investigated how markers of BC-1 and BC-2 behaved during repopulation of airway epithelium in these models of injury. We now show that the ventral and dorsal enrichment of these marker is remarkably maintained in these populations as the epithelium is undergoing repair. This further supports the idea that the regeneration-repair of the ventral and dorsal epithelium is mediated by BC-1 and BC-2-mediated programs of expansion and differentiation.

3. In some places the terminology chosen by the authors is confusing or unexplained. For example, the "signatures" of BC-1 and BC-2 "appeared to be the most stem-like BC populations" – it is not clear what the authors mean by "stem-like" and whether they are referring to specific markers (not mentioned) or functionality (not explicitly tested for stemness.) Consider explaining or deleting.

We referred to them as "the most stem-like” because in contrast to the other BC populations, the BC-1 and BC-2 were not characterized by enrichment in genes associated with differentiated (luminal) cellular phenotypes, such as Krt13 (Basal-squamous) or Secretoglobin and Mucin-related genes (Basal-secretory). Thus, while not based on enrichment for stem cell specific markers, BC-1 and BC-2 were characterized rather by exclusion of markers that identified the other BC populations. We have clarified this further in our revised manuscript.

4. The authors emphasize that dorsal and ventral basal cells are different because of different "niches", without studying what these niches or signals might be. Have the authors considered that the different structures, tensile/structural properties (e.g. stiffness, collapsibility) of the membranous (dorsal) trachea vs cartilaginous (ventral) trachea may contribute to epithelial intrinsic differences in cell behavior without differential niche signals. The removal of cartilage (Sox9 deletion studies in vivo) would then remove these structural differences by changing tensile properties of the local tissue (rather than removing a signaling niche), thus minimizing BC differences, as the authors observed in their knockout tracheas. Could the authors discuss or test for these alternative possibilities since conventional niche-stem cell signaling differences (e.g. "cartilage-derived signals") were not formally assayed in the present work to warrant such strong conclusions about distinct biological niches.

We thank the reviewer for reminding us about these important considerations, which are now included in the Discussion of our revised manuscript.

5. How different is the RA signaling activation between BC-1 vs. BC-2 or ventral vs. Dorsal regions? The authors demonstrated that RA signaling plays important roles in Krt13 expression by adding RA inhibitor, BMS, into ALI culture media, suggesting the importance of the niche environment for the determination of the cellular identities, BC-1 and BC-2. However, it is still unclear whether RA signaling is highly activated in BC-1 compared to BC-2. In addition, which niche cells such as chondrocytes or fibroblast secrete RA to BC-1 is unclear.

To address this question, we investigated sites of RA activation in the adult tracheal epithelium dung homeostasis using the RARE-LacZ reporter mice (Rossant et al., 1991). These mutants have been extensively shown to label cells activating RA signaling in various tissues and contexts. We compared the distribution of LacZ signals in the ventral and dorsal epithelium in sections of adult trachea co-stained with X-gal and markers of basal, secretory and multiciliated cells. Although LacZ signals were present in luminal (secretory, multiciliated) cells of both ventral and dorsal epithelium, we found no evidence of labeling in BCs (Figure 4—figure supplement 1). This suggests that, under homeostatic conditions, BCs are unlikely to activate RA signaling. However, the presence of components of the RA biosynthetic pathway in BCs suggests that they may produce RA or precursors making available to neighbor luminal cells to drive or maintain their differentiation program. Indeed in the human skin there is evidence that basal cells (keratinocytes) metabolize retinoid precursors and serve as a source of retinol for differentiation of the upper layers of skin (Kurlansky et al., J Biol. Chem. 1996).

Interestingly, BC-1 differs from BC-2 in its enrichment in gene components of the retinoid pathway or associated with it (KEGG: Retinol metabolism), suggesting that ventral (BC-1) may be better equipped of producing RA compared to dorsal (BC-2). This could potentially generate dorsal-ventral differences in sensitiveness of the tracheal epithelium to an RA-deficient environment manifested by the massive Krt13 induction in dorsal-derived BC cultures. Our previous studies in the developing lung showed that RA is very active early in the embryonic tracheal mesenchyme but later, during cartilage differentiation, signals already largely declined. (Malpel et al., Development 2000).

Reviewer #3 (Recommendations for the authors):The manuscript would benefit from some additional information/experiments.1) Material and methods seem to indicate that the basal cells isolated for organoid and ALI cultures were not sorted. Figure 2H shows some contamination with stromal cells, especially for the BC2 cultures. It is, therefore, possible that signals from these stromal cells are maintaining these BC1 vs BC2 phenotypes. To address this the authors may need to co-culture GFP positive BC1 isolated from a mouse in which all cells or basal cells are GFP + with Tomato positive BC2 isolated from a mouse in which all cells or basal cells are tomato +. Alternatively, basal cells could be isolated from mice in which Foxg1 positive cells express the Diphtheria toxin receptor so that stromal cells can be killed in culture.

This issue has been addressed in the current Figure 3—figure supplement 1 in which we show no evidence of contaminating Vimentin+ mesenchymal cells in these organoid cultures.

2) What are the different signals coming from the dorsal vs ventral stromal niches? Can the authors give some examples of differentially expressed genes for the different pathways described in Figure 1F?

Based on a database of genes differentially enriched in the cartilage primordia and smooth muscle of the E13.5 trachea, we speculate that potential signals in the dorsal vs ventral niches could be:

Wnt signaling:

Ventral: *Wnt5a, Notum, Axin2 Sfrp5 Barx1*

Dorsal: *Frzb Lgr6 Lgr5 Wnt4*

TGFB and Bmp signaling:

Ventral: *Msx2- Bmp4-Dlx1 Tgfbr2*

Dorsal: *Bmp7 Chrdl1 Bmp3*

Retinoic acid

Ventral: *Rarb*

3) Material and methods also describe the isolation of esophageal basal cells but this experiment does not seem to be included in this manuscript.

This has been corrected. We thank the reviewer and apologize for this oversight.

4) Where are the other 4 basal cell populations located?

As already indicated by our scRNAseq, these four populations of basal cells were relatively rare. We performed immunofluorescence staining with representative markers of these BC populations and now show that they are not associated with any specific location in the tracheal epithelium. The data are shown in new Figure 2—figure supplement 2.

5) This sentence "By contrast, BCs of the dorsal trachea were marked by strong expression of Cd44, Cav1, Krt17, which were only weakly expressed in ventral BCs (Figure 1G)." needs to be rephrased.

This sentence has been revised